# Autonomous Urban Region Representation with LLM-informed Reinforcement Learning

## Abstract

Urban representation learning has become a key approach for many applications in urban computing, but existing methods still rely heavily on manual feature designs and geographic heuristics. We present SubUrban, a reinforcement learning framework that autonomously discovers informative regional features through submodular rewards and semantic guidance from large language models. SubUrban adaptively expands each region into a hypernode, suppressing redundancy while preserving complementary associations, and learns cross-task embeddings with a graph-attention policy. Experiments across multiple prediction tasks (population, house price, and GDP) and cities (Beijing, Shanghai, New York, and Singapore) show that SubUrban consistently outperforms state-of-the-art baselines, achieving comparable accuracy with only 10% of the training data. These results highlight submodular-driven automation, enhanced by LLM-in-the-loop semantics, as a practical paradigm for autonomous urban region representation learning. The implementation of our SubUrban is available at
`https://anonymous.4open.science/r/SubUrban_ICLR2026`.

## 1 Introduction

Over the past decade, the rapid growth of large-scale urban data sources, including remote sensing imagery, points of interest (POIs), and human mobility records, has profoundly reshaped urban computing. These data provide unprecedented opportunities for *urban computing*, enabling applications in social analysis (Meyer & Turner, 1992), economic growth prediction (Hui et al., 2020), air quality modeling (Zheng et al., 2013), and traffic forecasting (Keller et al., 2020). Despite these advances, many approaches are tailored to specific tasks (Shimizu et al., 2021; Pulugurtha et al., 2013; Naik et al., 2014), require extensive labels, and cannot be readily adapted to other tasks.

Urban region representation learning (also called *urban region embedding*) has emerged as a promising approach to produce universal feature vectors of city regions that can be reused across tasks. The intuition is that urban applications often rely on common geospatial features. For instance, Wang et al. (Wang & Li, 2017) show that human mobility strongly correlates with socio-economic indicators such as crime rates, house prices, and household income. By embedding taxi trajectories into region representations, they achieved accurate predictions across diverse tasks. Building on this idea, subsequent studies generally combine two complementary perspectives: intra-region semantics and inter-region associations. Intra-region semantics characterize what is inside a region, such as building density, POI types, or land-use composition (Yuan et al., 2012; Zhang et al., 2017b; Yao et al., 2018; Fu et al., 2019; Zhang et al., 2019; 2020; Wang et al., 2020; Xi et al., 2022; Li et al., 2023; Huang et al., 2023; Balsebre et al., 2024). Inter-region associations describe how regions are related, for instance, through spatial proximity, functional similarity, or traffic connectivity (Wang & Li, 2017; Yao et al., 2018; Fu et al., 2019; Zhang et al., 2019; 2020; Wu et al., 2022; Zhang et al., 2022). These approaches reduce the cost of designing and training task-specific models.

Nevertheless, existing methods still demand significant human effort. For intra-region semantics, contrastive learning is widely used to highlight informative samples while suppressing noise, but it depends heavily on handcrafted geographic heuristics. For example, HGI (Huang et al., 2023) treats regions with moderately similar embeddings (cosine similarity 0.6–0.8) as hard negatives, while

RegionDCL (Li et al., 2023) selects regions with similar building clusters as positives to capture functional correlations. Such heuristics require domain expertise and suffer from costly preprocessing and training. For inter-region associations, researchers typically construct urban graphs where nodes represent POIs, buildings, or areas, and edges are derived from spatial distance, trajectories, or feature similarity. This giant design space makes it unclear which relations are most useful, often requiring extensive city-specific tuning and ad-hoc feature engineering. These challenges raise a central research question: **Can we design a framework that automatically identifies the most informative intra- and inter-region features to learn region embeddings, without relying on manual heuristics or city-specific adjustments?**

In this work, we identify two key challenges for automated region representation learning. First, prioritizing informative intra-region features requires domain knowledge, since not all input features are informative for downstream tasks, and real-world datasets often contain substantial redundancy and noise. For example, real-world POI datasets often contain large fractions of duplicated or low-informative entries such as addresses, phone numbers, or building facilities (e.g., block numbers, floor indices, elevators). Simply aggregating such entries not only increases computational overhead but also degrades the quality of learned embeddings. Second, the vast design space of graph structures makes it difficult to extract meaningful inter-region associations. Searching over possible urban graph constructions is both computationally expensive and challenging to optimize, as the number of candidate edges grows quadratically with the number of graph nodes.

To address these challenges, we propose **SubUrban**, a submodular-driven reinforcement learning framework for autonomous urban representation learning. SubUrban leverages submodular functions to suppress redundant POIs and prioritize informative features, while large language models provide city-specific heuristics to filter low-value data and highlight representative urban landmarks, enabling semantic-aware intra-region modeling. For inter-region relations, SubUrban applies submodular hypernode expansions that progressively connect each region to nearby and semantically complementary areas. This approach prunes the quadratic growth of candidate edges by retaining associations with the highest marginal utility. Experiments across multiple cities and tasks show that SubUrban outperforms state-of-the-art baselines with only 10% of the data, confirming the effectiveness of its redundancy suppression and semantic-aware selection strategies. To summarize, our contributions are at least threefold:

- We propose a novel **Sub**modular-driven reinforcement learning paradigm for autonomous **Urban** representation learning, eliminating the need for manual feature engineering and heuristic designs in data selection and region modeling.

- We introduce an LLM-informed framework that provides urban expertise and semantic guidance for informative candidate selection and exploration acceleration, enhancing both convergence efficiency and cross-city transferability.

- Extensive experiments demonstrate that SubUrban consistently outperforms state-of-the-art baselines across multiple tasks and cities, while achieving up to 90% data efficiency and robust transferability under diverse urban areas.

## 2 RELATED WORK

**Urban Region Representation Learning**  Early studies relied on task-specific features such as mobility patterns, social media check-ins, or remote sensing imagery for applications including air quality modeling, functional zone identification, and urban safety analysis (Yuan et al., 2012; Zheng et al., 2013; Yao et al., 2018; Naik et al., 2014). More recent work has shifted toward self-supervised paradigms that capture spatial correlations or inter-region interactions. Examples include flow-based embedding models (Wang & Li, 2017; Fu et al., 2019), proximity-constrained or contrastive approaches with graph encoders (Zhang et al., 2019; 2022), and multimodal fusion of text, imagery, and mobility signals (Jenkins et al., 2019; Zhang et al., 2017a; 2020; Wu et al., 2022). Extensions further incorporate heterogeneous data such as satellite imagery and building footprints (Li et al., 2023; Huang et al., 2023; Balsebre et al., 2024; Yan et al., 2024; Wang et al., 2020). While these methods significantly improve reusability across tasks, they still depend on heuristic choices for sample construction and city-specific tuning, and often suffer from redundancy when large-scale urban data are indiscriminately included.

**LLMs for Urban Tasks**  Large Language Models (LLMs) have recently been applied to urban computing for their ability to enrich semantics and contextual reasoning. Representative directions include domain adaptation for geoscientific corpora (Deng et al., 2024), LLM-guided region descriptions (Fu et al., 2024), and LLM-agent frameworks for building urban knowledge graphs and aligning heterogeneous sources (Ning & Liu, 2024; Manvi et al., 2024). These works highlight the potential of LLMs in urban data mining, but their role in guiding representation learning remains underexplored, particularly in evaluating and prioritizing informative regional features.

## 3 PRELIMINARIES

**Definition 1** (Urban Hypernode). *An urban hypernode $\mathcal{S}_r$ is an extended representation unit that includes both POIs within a region $r$ and selected POIs from its $\delta$-neighborhood. Given candidates $\mathcal{P}_r = \{p \in POI \mid dist(p, r) \leq \delta_r\}$, a subset $\mathcal{S}_r \subseteq \mathcal{P}_r$ is chosen based on spatial structure, semantic relevance, and submodular rewards. The resulting hypernode $(r, \mathcal{S}_r)$ enriches region representation with contextual information beyond the boundary.*

**Definition 2** (Urban Region Representation Learning). *Given regions $\mathcal{U} = \{u_1, \dots\}$, the goal is to learn a mapping that produces a vector $\mathbf{z}_i \in \mathbb{R}^d$ for each $u_i \in \mathcal{U}$, which can be used in downstream prediction tasks such as population density or housing price prediction.*

**Definition 3** (Submodular Reinforcement Learning). *Submodular reinforcement learning models rewards as submodular set functions to capture diminishing returns. For a ground set $V$, a function $F : 2^V \to \mathbb{R}$ is submodular if*

$$F(A \cup \{v\}) - F(A) \geq F(B \cup \{v\}) - F(B) \tag{1}$$

*for all $A \subseteq B \subseteq V$ and $v \in V \setminus B$.*

**Problem Statement.**  Given a set of urban regions $\mathcal{R} = \{r_1, \dots, r_n\}$ with their surrounding POI distributions, our goal is to learn an adaptive expansion policy network that constructs urban hypernodes for optimal region representation. Formally, we aim to optimize:

$$\pi_\theta^* = \arg\max_{\pi_\theta} \mathbb{E}_{r \sim \mathcal{R}} \left[ R\left((r, \pi_\theta(\mathcal{P}_r)), \mathcal{T}\right) \right] \tag{2}$$

where $\pi_\theta : \mathcal{P}_r \to \mathcal{S}_r \subseteq \mathcal{P}_r$ represents the expansion policy network that selects POI subset $\mathcal{S}_r$ from the candidate set $\mathcal{P}_r$, and $R(\cdot, \mathcal{T})$ denotes the reward function evaluated on downstream tasks $\mathcal{T}$. It is noteworthy that we focus on POIs in this work since they are the most widely used features in literature (Chen et al., 2024); however, the framework is general and can be extended to other textual inputs or adapted to visual modalities via vision–language models.

## 4 METHODOLOGY

We present the SubUrban framework, which comprises three key components as illustrated in Figure 1: (1) **POI Set Preprocessing** applies LLM-guided semantic retrieval and spatial clustering to condense raw POI data while preserving structural diversity and functional relevance. (2) **Submodular-Aware Reinforcement Learning** formulates POI selection as a sequential decision task, where an agent selects POIs based on submodular utility within structured spatial contexts. (3) **LLM-Instructed CEM Optimization** calibrates attention weights of POI categories based on heuristics from LLM instruction to enhance semantic alignment and accelerate convergence.

### 4.1 POI SET PREPROCESSING

Urban data such as POIs, check-ins, and geo-tagged tweets are often massive, noisy, and redundant. Existing approaches either manually curate a limited set of useful inputs, which requires significant human labor and domain knowledge, or simply feed all available data into training, which increases computation and amplifies noise. To address these problems, we adopt a more selective strategy with LLMs. Instead of feeding all candidate POI to LLMs, which would be prohibitively costly and slow, we only provide the administrative region's name and address,  prompting it to generate heuristic keywords. For well-known regions, the model tends to return landmarks and attractions (e.g., Times Square), while for less prominent regions it generates important functional roles (e.g., residential

Reviewer D3YD-W1

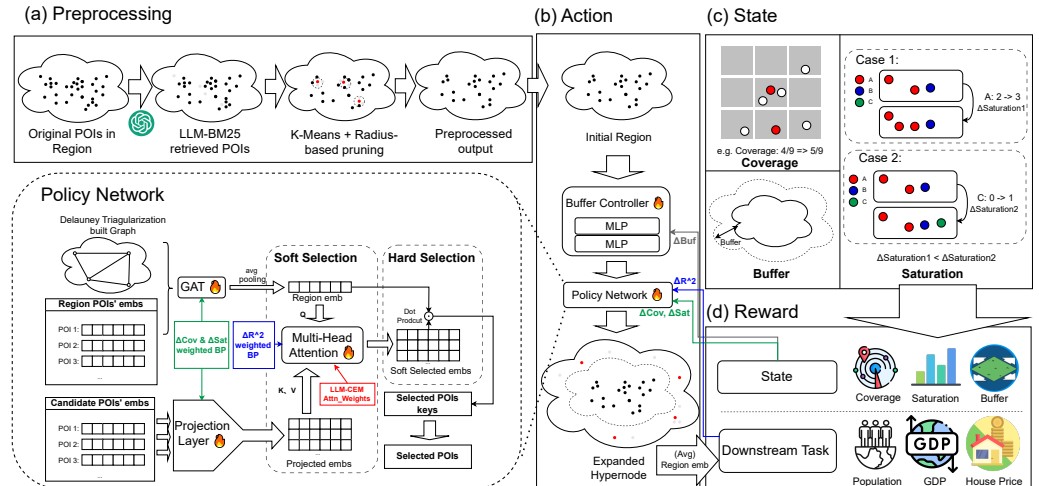

Figure 1: Overview of the SubUrban learning framework. With a defined triplet (Coverage, Saturation, Buffer) as the State, delta values of mixed downstream results and states as Reward, a two-stage policy network as Action to extend POI.

or industrial) of the area. We then apply off-the-shelf retrieval methods (e.g., BM25 (Robertson & Walker, 1994)) to locate POIs that match these keywords. Next, K-means clustering is applied to regulate spatial density and ensure more uniform coverage across the administrative regions, resulting in a functionally representative subset of POIs, which can serve as reliable starting points for further expansion.

Reviewer D3YD-W1

## 4.2 SUBMODULAR-AWARE REINFORCEMENT LEARNING

To automate the process of identifying informative intra- and inter-region features, we mimic how human experts gradually refine their understanding of a city. Rather than fixing rules in advance, experts iteratively select features, evaluate their usefulness based on domain-specific criteria or validation tasks, and adjust their choices accordingly. This adaptive trial-and-error process is naturally aligned with reinforcement learning, formalized by a three-tuple (state, action, reward). In our setting, these are defined as geospatial states, feature-selection actions, and submodular-aware rewards.

### 4.2.1 GEOSPATIAL-DEFINED STATE

We define the **state** to capture the properties of currently selected POIs, summarizing their spatial extent (**Coverage, Cov**), semantic diversity (**Saturation, Sat**), and potential for future expansion (**Buffer, Buf**). Intuitively, each POI represents certain urban functions within its surrounding area[1]. The buffer component (**Buf**) is inspired by previous submodular RL work (Prajapat et al., 2024), reflecting the fact that adding more data points beyond a certain level brings diminishing returns. Once key urban functions are sufficiently represented, further expansion offers little additional benefit.

Formally, we represent the state as a triplet $\text{State}_t = (Cov_t, Sat_t, Buf_t)$, where

$$Cov_t = \frac{|\{g : g \cap S_t \neq \emptyset\}|}{|\mathcal{G}|}, \quad Sat_t = -\frac{1}{\log C}\sum_c q_c \log q_c, \quad Buf_t = f_{\text{MLP}}(\text{State}_{t-1}). \quad (3)$$

Here, $Cov_t$ denotes the proportion of grid cells already covered by selected POIs from the current selection set $S_t$, where $\{g : g \cap S_t \neq \emptyset\}$ represents the set of grid cells that intersect with at least one POI in $S_t$. $Sat_t$ is the normalized entropy of POI category distribution, where $q_c$ is the proportion of POIs belonging to category $c$ and $C$ denotes the total number of POI categories . $Buf_t$ is an adaptive expansion radius predicted by a two-layer MLP with softplus activation to control how far new candidates are retrieved at step $t$.

Reviewer D3YD-W2

---

[1]The intuition is general and can be adapted to other urban data types like buildings and street-view images.

### 4.2.2 GEOSPATIAL-BASED ACTION

In the SubUrban framework, the action represents how the system autonomously expands the regional POI set to construct more informative hypernodes. Because neither human experts nor LLMs can exhaustively examine city-scale data, we mimic the strategy of human experts who first conduct fine-grained sensing to capture potentially useful information, and then apply a unified standard to filter the data. Following this intuition, our policy alternates between soft selection, which preserves recall through attention-based scoring of candidate POIs, and hard selection, which contracts the set by dot-product similarity to produce a compact and representative subset.

**Soft Selection**  We assess the importance of candidate POIs by evaluating how their features contribute to the aggregated region embeddings. Following the definitions in Eq. 4, $\mathbf{p}_j$ denotes the embedding of an intra-region POI from $S_r$, while $\mathbf{p}_i$ refers to the embedding of a candidate POI drawn from the buffer set $B_r$. We encode intra-region POIs with a Graph Attention Network (GAT) using Delaunay triangulation (Delaunay, 1934) edges $\mathcal{E}_r$ following Huang et al. (2023); Balsebre et al. (2024); Li et al. (2023), and apply average pooling to obtain the region embedding and compute candidate importance in a single step:   Reviewer D3YD-W2

$$\mathbf{z}_r = \frac{1}{|S_r|} \sum_{j \in S_r} \mathrm{GAT}(\mathbf{p}_j, \mathcal{E}_r), \quad \alpha_i = \frac{1}{H} \sum_{h=1}^{H} \mathrm{Attn}(\mathbf{z}_r, W_P \mathbf{p}_i). \tag{4}$$

The scores are then reweighted by category weights $w_{c(i)}$ from the LLM-instructed CEM process in Section 4.3, and candidates and their associated edges are retained only if their weighted scores exceed the threshold, with an additional cap of $K_{\mathrm{soft}}$ to prevent oversampling in dense regions.

$$\tilde{\alpha}_i = \alpha_i \cdot w_{c(i)}, \quad \bar{\tilde{\alpha}} = \frac{1}{|\mathcal{B}_r|} \sum_{j \in \mathcal{B}_r} \tilde{\alpha}_j, \quad \mathcal{S}_r^{\mathrm{soft}} = \mathrm{Top}_{K_{\mathrm{soft}}}\{\mathbf{p}_i \in \mathcal{B}_r \mid \tilde{\alpha}_i \geq \bar{\tilde{\alpha}}\}. \tag{5}$$

**Hard Selection**  To obtain a compact and consistent subset, we refine the soft candidates by dot-product similarity to the regional embedding $\mathbf{z}_r$. Each similarity score is reweighted by the same category preferences and compared against the mean score $\bar{\mathbf{s}}$. Only candidates above this threshold are retained, subject to a cap $K$ that prevents oversampling in dense regions:

$$\mathcal{S}_r^{(t)} = \Big\{ \mathbf{p}_i \in \mathcal{S}_r^{\mathrm{soft}} \,\big|\, w_{c(i)} \cdot (\mathbf{z}_r^\top W_P \mathbf{p}_i) \geq \bar{\mathbf{s}}, \, |\mathcal{S}_r^{(t)}| \leq K \Big\}. \tag{6}$$

Here $\bar{\mathbf{s}}$ denotes the mean of all weighted similarity scores within $\mathcal{S}_r^{\mathrm{soft}}$. This step contracts the candidate pool into a smaller yet representative subset, and we set $K_{\mathrm{soft}} = 1.5K$ for simplicity. The final number of expanded POIs for each region, denoted as the dynamic $\delta_r$, is obtained directly in this hard selection stage by keeping only the candidates whose similarities exceed the mean threshold. The size of the remaining set naturally becomes the value of $\delta_r$.   Reviewer D3YD-W1

The soft and hard selections are executed alternately, removing redundant POIs and edges while preserving informative ones. With the guidance of the reward signals, it progressively shapes more coherent submodule structures across regions, thereby capturing useful inter-region relationships.

### 4.2.3 REWARD AND LOSS FUNCTIONS

**Reward Function**  Different modules in SubUrban focus on different aspects of the learning process, so we design tailored reward signals rather than a single global metric. Intuitively, the GAT and projection layers should capture local improvements in spatial coverage and semantic diversity, the attention module should be aware of the global task performance, and the buffer controller should balances task performance with expansion constraints. We define three reward signals corresponding to the GAT, Projection Layer, Multhead Attention, and buffer controller modules:

$$R_{\mathrm{GAT}} = R_{\mathrm{proj}} = \frac{\Delta_{\mathrm{sat}}}{\sigma_{\mathrm{sat}}} + \frac{\Delta_{\mathrm{cov}}}{\sigma_{\mathrm{cov}}}, \tag{7}$$

$$R_{\mathrm{MHA}} = \frac{\Delta_{\mathrm{downstream}}}{\sigma_{\mathrm{downstream}}}, \tag{8}$$

$$R_{\mathrm{buf}} = \frac{\Delta_{\mathrm{downstream}}}{\sigma_{\mathrm{downstream}}} + \alpha_{\mathrm{buf}} \cdot \frac{\Delta_{\mathrm{buf}}}{\sigma_{\mathrm{buf}}} - \max(\Delta_{\mathrm{buf}} - \beta_{\mathrm{buf}} \cdot \mathrm{Buf}_t, 0). \tag{9}$$

Here $R_{\text{GAT}}$ and $R_{\text{proj}}$ guide the GAT and Projection Layer using local state signals, i.e., improvements in semantic diversity ($\Delta_{\text{sat}}$) and spatial coverage ($\Delta_{\text{cov}}$). $R_{\text{MHA}}$ directs the Multihead Attention using improvements in downstream task performance on the validation set ($\Delta_{\text{downstream}}$). $R_{\text{buf}}$ steers the buffer controller by combining downstream performance with expansion constraints, controlled by $\alpha_{\text{buf}}$ and $\beta_{\text{buf}}$ (with sensitivity analyzed in Appendix D.11). All reward terms are normalized by historical standard deviations ($\sigma_{\text{sat}}, \sigma_{\text{cov}}, \sigma_{\text{downstream}}, \sigma_{\text{buf}}$) to stabilize scales without manual tuning.

**Advantage Function.** To reduce reward variance and stabilize training across modules, each module maintains an Exponential Moving Average (EMA) that tracks the expected reward over time. The advantage function for continuous-time reinforcement learning (Baird, 1994) is applied to compute the difference between the current reward and EMA, providing a normalized signal that indicates whether the current performance exceeds historical expectations:

$$A_t^{(m)} = R_t^{(m)} - b_t^{(m)}, \quad b_t^{(m)} = \gamma_t \cdot b_{t-1}^{(m)} + (1 - \gamma_t) \cdot R_t^{(m)}, \quad \gamma_t = \sigma\left(\frac{\|R_{t-1}^{(m)} - R_{t-2}^{(m)}\|}{\|R_{t-1}^{(m)}\| + \epsilon}\right) \quad (10)$$

where $(m)$ can be modules including GAT, Projection Layer, Multihead Attention, and Buffer Controller. And $\gamma_t$ is the adaptive EMA coefficient computed from reward variability when sufficient training history becomes available, eliminating manual parameter tuning.

**Loss and Gradient Updates** Each module employs advantage-weighted gradient updates with distinct optimization strategies tailored to its specific learning objectives. The Buffer Controller uses PPO-style clipped ratios for stability, the Multihead Attention mechanism applies cross-entropy loss weighted by mixed-task advantages to optimize selection quality, while GAT and Projection Layer directly use advantage-weighted updates to optimize state representation quality:

$$\nabla_{\theta_{\text{buf}}}\mathcal{L}_{\text{buf}} = -\mathbb{E}\left[\min(r_t A_t^{\text{buf}}, \text{clip}(r_t, 1 - \epsilon, 1 + \epsilon) A_t^{\text{buf}})\right] \quad (11)$$

$$\nabla_{\theta_{\text{MHA}}}\mathcal{L}_{\text{MHA}} = -\mathbb{E}\left[\mathcal{L}_{\text{cross-entropy}} \cdot A_t^{\text{MHA}}\right] \quad (12)$$

$$\nabla_{\theta_{\text{GAT,proj}}}\mathcal{L}_{\text{GAT,proj}} = -\mathbb{E}\left[A_t^{\text{GAT,proj}}\right] \quad (13)$$

where $r_t$ is the ratio of action probabilities under the updated and previous buffer policies following PPO settings (Schulman et al., 2017).

## 4.3 LLM-INSTRUCTED CEM OPTIMIZATION

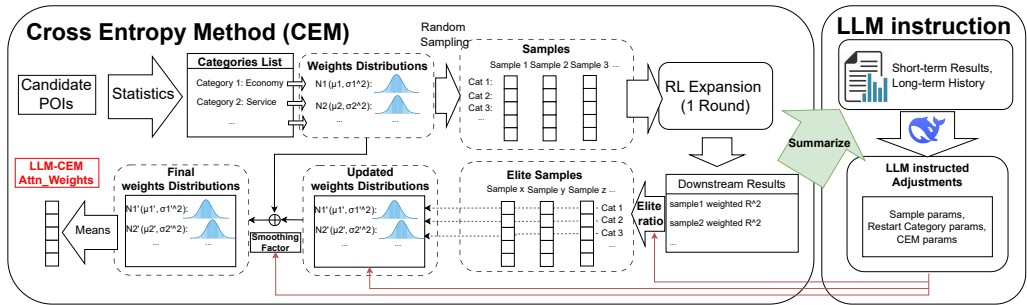

Figure 2: LLM-instructed CEM tunes category weights of POIs in the Multihead Attention module from the policy network.

Human-designed heuristics have proven effective in previous studies Chen et al. (2024), but they demand costly manual effort. In SubUrban, we instead use a Large Language Model to inject heuristics automatically. The LLM can continuously observe the evolving optimization process, improving the selection and accelerating the convergence via its feedback. Specifically, we initialize the category weights $\{w_c\}_{c=1}^C$ that scale the attention scores in Eq. 4 via Cross Entropy Method (CEM), which iteratively samples candidate weight vectors from Gaussian distributions with per-category means $\mu_c^{(t)}$ and standard deviations $\sigma_c^{(t)}$, selects "elite samples" based on downstream task performance,

and updates the distribution toward samples with high performance as follows:

$$\mu_c^{(t+1)} = \alpha \, \mu_c^{(t)} + (1 - \alpha) \, (\mu_{\text{elite}}^{(t)})_c, \quad \sigma_c^{(t+1)} = \alpha \, \sigma_c^{(t)} + (1 - \alpha) \, (\sigma_{\text{elite}}^{(t)})_c \tag{14}$$

Then, we utilize the Large Language Model to analyze optimization behavior and provide targeted parameter adjustments. The LLM observes both recent optimization behavior and long-term history, and proposes heuristic adjustments to distribution parameters and stability factors. This high-level guidance complements the sampling-driven updates of CEM, while detailed interaction protocols and implementation settings are deferred to Appendix C.2 and Appendix B.

## 5 EXPERIMENTS

In this section, we evaluate the proposed method and the derived representation of extended POI subsets following previous literature (Li et al., 2023; Balsebre et al., 2024). We also perform ablation studies, case studies, and parameter sensitivity analysis.

### 5.1 EXPERIMENTAL SETTINGS

**Dataset** We conduct experiments using inputs of POI datasets collected via the Gaode Map API for Beijing and Shanghai, while from OSM for Singapore and New York City. The statistics of POI datasets are shown in Table 1.

Table 1: Dataset Statistics

| City | POIs | POI categories | Regions |
|------|------|----------------|---------|
| Beijing | 1,218,188 | 23 | 1,253 |
| Shanghai | 1,192,123 | 22 | 1,688 |
| Singapore | 269,961 | 759 | 2,520 |
| New York City | 283,810 | 65 | 2,280 |

**Baselines and Metrics** We compare SubUrban against seven urban region representation learning baselines through POI encoding: BERT (Devlin et al., 2019a), OpenAI (Neelakantan et al., 2022), GraphSage (Hamilton et al., 2017), DGI (Zhao et al., 2023), MVGRL (Hassani & Ahmadi, 2020), HGI (Huang et al., 2023), and CityFM (Balsebre et al., 2024), with details in Appendix D.1. We focus on POI encoding methods since our approach addresses diminishing returns from POI data specifically. Evaluation is conducted on three regression tasks: population density prediction, house price prediction, and GDP density prediction using the classifier of Random Forest with 4:1 train/test splits. We report the performance through average and standard deviation across 5 runs with different random seeds under 5-fold cross-validation, using Mean Absolute Error (MAE), Root Mean Squared Error (RMSE), and Coefficient of Determination (R²) metrics.

### 5.2 EXPERIMENTAL RESULTS

We evaluate the quality of derived representations from our proposed SubUrban and other baselines from cross-city and cross-task aspects.

#### 5.2.1 CROSS-CITY PERFORMANCE

We conduct population density prediction experiments across four diverse cities (Beijing, Shanghai, Singapore, and New York City) to evaluate the cross-city adaptability of SubUrban. Table 2 shows that graph structural methods (GraphSAGE, DGI, MVGRL) exhibit inconsistent performance among different cities, suggesting that differences in urban planning contexts affect the effectiveness of graph learning. Strong baselines incorporating both semantic and spatial contexts (HGI, CityFM) achieve more consistent results across cities, with CityFM demonstrating the best baseline performance through extensive OpenStreetMap pretraining. SubUrban outperforms all baselines across all four cities, demonstrating cross-city adaptability using only 10% of the full POI set.

#### 5.2.2 CROSS-TASK PERFORMANCE

We extend the evaluation to house price and GDP density prediction in Beijing to evaluate the cross-task adaptability of SubUrban. Table 3 shows that graph structural methods (GraphSAGE, DGI, MVGRL) exhibit inconsistent performance across cities and tasks, often comparable to simple averaging (BERT-Avg). These methods show better performance in Singapore and NYC compared to

Table 2: Population Density Prediction in Beijing, Shanghai, Singapore, and NYC Reviewer D3YD-W3

| Models | Beijing | | | Shanghai | | | Singapore | | | NYC | | |
|---|---|---|---|---|---|---|---|---|---|---|---|---|
| | MAE↓ | RMSE↓ | $R^2$↑ | MAE↓ | RMSE↓ | $R^2$↑ | MAE↓ | RMSE↓ | $R^2$↑ | MAE↓ | RMSE↓ | $R^2$↑ |
| BERT-Avg | 5043.73 (±170.77) | 8203.42 (±198.00) | 0.49 (±0.02) | 9375.19 (±75.37) | 14235.93 (±203.54) | 0.47 (±0.01) | 4002.01 (±206.71) | 5818.71 (±329.95) | 0.68 (±0.01) | 5325.16 (±129.20) | 6845.95 (±159.71) | 0.56 (±0.02) |
| OpenAI-Avg | 5419.69 (±158.87) | 8440.61 (±172.59) | 0.46 (±0.02) | 9816.80 (±108.09) | 14579.09 (±305.37) | 0.44 (±0.02) | 3896.15 (±85.61) | 5657.27 (±89.50) | 0.69 (±0.03) | 3858.68 (±102.36) | 5366.48 (±201.16) | 0.73 (±0.02) |
| GraphSage | 4774.99 (±269.06) | 7812.94 (±578.01) | 0.52 (±0.07) | 8759.62 (±388.16) | 13682.10 (±644.79) | 0.53 (±0.02) | 3424.71 (±117.11) | 5280.75 (±206.49) | 0.74 (±0.02) | 4025.13 (±95.00) | 5502.47 (±140.00) | 0.72 (±0.02) |
| DGI | 4990.86 (±150.99) | 8153.15 (±522.52) | 0.47 (±0.07) | 9315.73 (±441.15) | 14110.26 (±1157.06) | 0.47 (±0.05) | 3925.79 (±206.24) | 5720.04 (±385.50) | 0.73 (±0.03) | 4291.90 (±78.71) | 5847.58 (±110.30) | 0.69 (±0.01) |
| MVGRL | 4990.86 (±150.99) | 8153.15 (±522.52) | 0.47 (±0.07) | 9087.51 (±573.17) | 13646.88 (±1078.60) | 0.49 (±0.06) | 4014.24 (±301.56) | 5932.78 (±542.86) | 0.70 (±0.03) | 4693.77 (±75.48) | 6414.38 (±115.52) | 0.62 (±0.01) |
| HGI | 4534.83 (±473.15) | 7446.83 (±746.63) | 0.56 (±0.09) | 7464.74 (±182.11) | 11642.35 (±289.60) | 0.66 (±0.02) | 3393.52 (±216.56) | 5035.43 (±295.80) | 0.76 (±0.02) | 3957.31 (±46.34) | 5424.56 (±158.39) | 0.72 (±0.02) |
| CityFM | 4199.19 (±65.02) | 6858.44 (±143.30) | 0.64 (±0.02) | 6558.20 (±108.37) | 10677.55 (±218.36) | 0.71 (±0.01) | 3085.52 (±104.42) | 4504.32 (±203.52) | 0.82 (±0.01) | 3697.40 (±122.25) | 5243.60 (±196.12) | 0.74 (±0.02) |
| **SubUrban** | **3283.11** (±273.61) | **5719.89** (±640.22) | **0.72** (±0.06) | **5684.80** (±356.93) | **9673.78** (±716.99) | **0.75** (±0.02) | **2475.59** (±180.29) | **4266.60** (±455.03) | **0.86** (±0.03) | **3401.17** (±167.26) | **4937.25** (±245.88) | **0.77** (±0.02) |

Beijing and Shanghai, suggesting urban planning differences affect effectiveness. Strong baselines (HGI, CityFM) demonstrate more consistent results, with CityFM achieving superior population prediction through OpenStreetMap pretraining, while HGI shows stronger house price prediction via rule-based negative sampling. SubUrban consistently outperforms all baselines across all tasks, especially achieving notable improvements in Population Density and House Price prediction tasks, which demonstrates the cross-task adaptivity.

Table 3: Population Density, House Price, and GDP Density Prediction in Beijing

| Models | Population | | | House Price | | | GDP Density | | |
|---|---|---|---|---|---|---|---|---|---|
| | MAE↓ | RMSE↓ | $R^2$↑ | MAE↓ | RMSE↓ | $R^2$↑ | MAE↓ | RMSE↓ | $R^2$↑ |
| BERT-Avg | 5043.73 (±170.77) | 8203.42 (±198.00) | 0.49 (±0.02) | 14391.39 (±681.11) | 20622.46 (±785.23) | 0.74 (±0.03) | 490.47 (±36.14) | 789.56 (±62.61) | 0.62 (±0.04) |
| OpenAI-Avg | 5419.69 (±158.87) | 8440.61 (±172.59) | 0.46 (±0.02) | 13946.38 (±695.83) | 20105.17 (±1155.70) | 0.75 (±0.03) | 523.66 (±42.84) | 815.47 (±67.57) | 0.59 (±0.03) |
| GraphSage | 4774.99 (±269.06) | 7812.94 (±578.01) | 0.52 (±0.07) | 14748.74 (±2750.97) | 22275.26 (±5175.66) | 0.69 (±0.17) | 488.91 (±36.89) | 782.01 (±89.28) | 0.63 (±0.04) |
| DGI | 4990.86 (±150.99) | 8153.15 (±522.52) | 0.47 (±0.07) | 15357.90 (±1876.32) | 20122.38 (±3558.96) | 0.75 (±0.06) | 466.77 (±23.28) | 743.05 (±74.46) | 0.67 (±0.05) |
| MVGRL | 4990.86 (±150.99) | 8153.15 (±522.52) | 0.47 (±0.07) | 15692.40 (±1534.83) | 22317.73 (±3920.51) | 0.70 (±0.04) | 502.90 (±25.72) | 840.42 (±69.60) | 0.57 (±0.07) |
| HGI | 4534.83 (±473.15) | 7446.83 (±746.63) | 0.56 (±0.09) | 14719.13 (±1378.46) | 19008.63 (±1834.69) | 0.78 (±0.05) | 409.07 (±34.54) | 695.99 (±69.44) | 0.70 (±0.02) |
| CityFM | 4199.19 (±65.02) | 6858.44 (±143.30) | 0.64 (±0.02) | 14291.54 (±371.40) | 19483.32 (±582.32) | 0.75 (±0.02) | 384.27 (±18.37) | 601.26 (±48.58) | 0.78 (±0.04) |
| **SubUrban** | **3283.11** (±273.61) | **5719.89** (±640.22) | **0.72** (±0.06) | **12235.97** (±1249.12) | **17021.29** (±2364.06) | **0.85** (±0.03) | **349.63** (±27.50) | **568.85** (±42.24) | **0.80** (±0.03) |

### 5.2.3 EFFICIENCY ANALYSIS

We compare the Total Processing Time in minutes between our proposed SubUrban and strong baselines (CityFM and HGI). The total processing time includes the time of data preprocessing, model training, and encoding with evaluations. SubUrban achieves the shortest processing time with the highest performance among strong baselines as it utilizes LLMs to efficiently filter out noise POIs and accelerate convergence, while baselines spend much time on training with redundant POIs.

Table 4: Total Processing Time (Minutes)

| Method | Beijing | Shanghai |
|---|---|---|
| CityFM | 535 | 609 |
| HGI | 2262 | 3790 |
| SubUrban | 375 | 395 |
| Saves (%) | 29.9% | 35.1% |

## 5.3 ABLATION STUDIES

**Impact of Model Components**   We validate the effect of key components within SubUrban by comparing with the following variants: **w/o RL**: excludes RL-driven expansion, using random expansion instead; **w/o CEM**: omits LLM-instructed CEM optimization; **Ours**: the complete SubUrban framework. The results in Figure 3 demonstrate that both components significantly enhance prediction performance across all metrics. The absence of RL leads to the most substantial performance decline, with MAE increasing by approximately 8-12% for population prediction and 4-8% for house price prediction across both cities, confirming that intelligent RL-driven expansion is crucial for capturing optimal spatial patterns. Removing CEM optimization also degrades performance with consistent drops of 2-4% across all metrics. Additionally, we evaluate the LLM instruction by comparing LLM preprocessed POIs against random sampling and LLM-instructed CEM against pure CEM optimization, finding consistent improvements in training convergence and reward curves (details in Appendix D.10.4, and more studies in Appendix D.12).

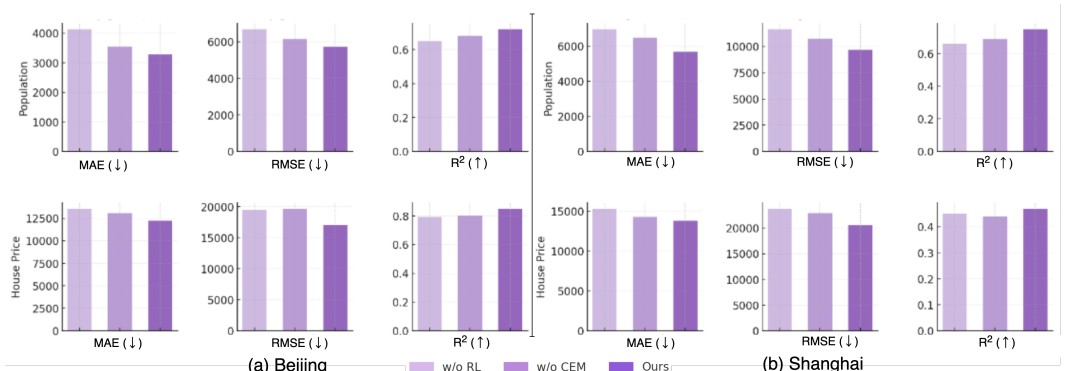

Figure 3: Ablation results of Population Density (first row) and House Price prediction.

**Impact of Data-sparsity**   Since urban data is unevenly distributed in space, we evaluate how SubUrban adapts to POI-sparse regions. We partition all regions into four groups of equal size (328 each) based on POI counts and report the MAEs of both tasks in Figure 4. The results show that SubUrban consistently achieves the lowest prediction errors across all density levels, and remains superior to baselines even when using only 10% of the full POI set. This robustness to data sparsity suggests that SubUrban can generalize better across cities with varying information densities.

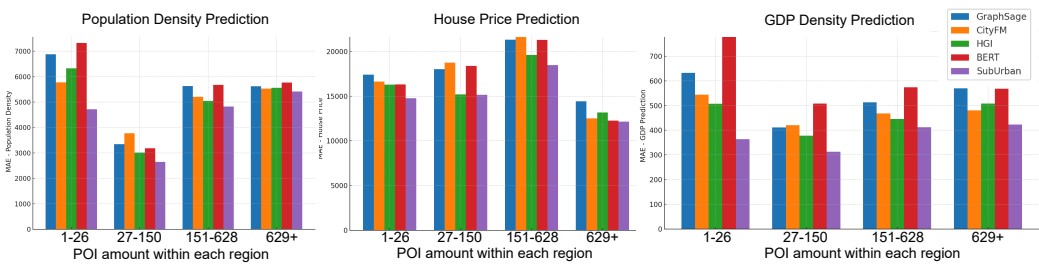

Figure 4: Mean Absolute Error (MAE) of Prediction tasks in regions with different numbers of POIs in Beijing.

**Parameter Sensitivity Analysis**   We further analyze the sensitivity of SubUrban on the penalty coefficient $\alpha$ and the Top-K, where SubUrban achieves stable performance across wide ranges of both parameters, indicating robustness and reducing the need for expert tuning. Due to the page limit, we present the details in Appendix D.11.

# 6 CONCLUSION

In this work, we propose SubUrban, a submodular-aware reinforcement learning framework for urban region representation, focusing on automatically identifying POIs that maximize informativeness and adaptivity while minimizing redundancy. By jointly modeling coverage, saturation, and buffer through a hypernode expansion process, SubUrban adaptively prioritizes spatially and semantically complementary POIs while mitigating redundancy from start to convergence, enabling effective selection and efficient optimization under the vast design space. Experiments on cross-city and cross-task comparison demonstrate superior performance over strong baselines with up to 90% less data, robustness across varying POI densities, and insensitivity with respect to buffer distance and candidate set size. This study establishes a new paradigm of autonomous urban representation learning, offering a transformative framework across cities and tasks with improved robustness, transferability, and data efficiency.

## ETHICS STATEMENT

We leverage publicly available and non-identifiable data sources, including POI datasets collected from Gaode Map API for Beijing and Shanghai, and OpenStreetMap for Singapore and New York City. All datasets contain only aggregated place-level information without any personal identifiers. No individual-level mobility records or sensitive demographic data are used. Our proposed framework focuses on urban region representation learning with the aim of improving predictive modeling of population density, house prices, and GDP density at the regional level. Our methodology cannot be used to identify or track specific individuals.

## REPRODUCIBILITY STATEMENT

We have made careful efforts to ensure the reproducibility of our work. The overall framework design, including the submodular-driven reinforcement learning formulation and the hypernode expansion process, is described in Section 4.2. Implementation settings are reported in Appendix B. Dataset statistics and sources are presented in Table 1 and Table 5. Finally, we will release anonymized source code through the link at the end of the Abstract to facilitate independent verification and reproduction of experiments.

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

## A  DATA SOURCE

In this paper, all datasets we used are available online. We hereby provide their links in Table 5.

Table 5: Data sources and links

| Data Type | Source | Link |
|---|---|---|
| POI datasets - Bejing, Shanghai | Gaode - API search | `https://lbs.amap.com/` |
| POI datasets - Singapore, NYC | OpenStreetMap | `https://download.geofabrik.de/` |
| Region partitions - Beijing, Shanghai | GADM | `https://gadm.org` |
| Region partitions - Singapore | OpenStreetMap | OSM Overpass API |
| Region partitions - NYC | NYC Planning | `https://www.nyc.gov/content/planning/pages/` |
| Population density | WorldPop | `https://hub.worldpop.org` |
| House prices | Beike | `https://ke.com` |
| Gross Domestic Product (GDP) | RESDP | `https://doi.org/10.12078/2017121102` |

## B  IMPLEMENTATION DETAILS

Due to the original settings of different baselines, the dimension $d$ of generated representation varies. The dimension $d = 768$ for BERT, $d = 1536$ for OpenAI, $d = 64$ for HGI, $d = 512$ for DGI and MVGRL, $d = 1024$ for CityFM and GraphSAGE. For our SubUrban, the average pooling of hypernode subset results in the same output dimension $d = 768$ as the BERT embeddings of POIs in regions. We set 15 rounds of CEM optimization with early stopping, 10 rounds of RL expansion for both training and testing phases of SubUrban, with a training set comprising only 1/10 of regions that provide downstream feedback. All experiments are conducted on 1 NVIDIA V100 32 GB GPU unit.

## C  LLM INSTRUCTIONS

### C.1  LLM INSTRUCTIONS ON POI PRE-SELECTION

To prepare the POI set for the start point of SubUrban, which is also the POI preprocessing step with LLM knowledge mentioned in Section 4.1, we generated representative keywords for each administrative region using GPT4. In the case of New York City, the five Boroughs (*Manhattan*, *Brooklyn*, *Queens*, *Bronx*, *Staten Island*) are the administrative regions.                    Reviewer D3YD-W1

**Prompt Template.**    We use GPT-4 to produce the representative keywords for each Borough in
NYC. The user prompt specified categories such as landmarks, shopping centers, transportation    Reviewer D3YD-W1
hubs, cultural venues, residential or neighborhood features, businesses, historical sites, and major
districts. The template of the prompt is as follows:

```
For the following NYC borough:  <borough_name>

Please generate a concise set of representative keywords
that capture the essential characteristics and features
of this borough.  The following categories are provided
solely as non-exhaustive illustrative examples to guide the
generation of relevant keywords:

- Notable landmarks, buildings, or attractions (e.g.,
museums, parks, iconic buildings)
- Shopping centers, markets, or commercial districts
- Transportation hubs (subway stations, bridges, major
streets)
- Cultural institutions or entertainment venues
- Residential developments, housing projects, or
neighborhood characteristics
- Local businesses, restaurants, or community features
```

```
            - Historical sites or points of interest
            - Major neighborhoods or districts within the borough

            Provide the keywords in a comma-separated format within
            single quotes, as in:  'keyword1','keyword2','keyword3',...
```

**Output Format**    The final output for each Borough was stored in a tab-separated format as follows:

```
            BOROUGH_NAME    'keyword1','keyword2',...
```

## C.2   LLM INSTRUCTIONS ON CEM

As mentioned in Section 4.3, we use LLM to instruct the CEM process for a faster convergence of
the optimal category weights searching process.

**Prompt Template.**   We use GPT-4 to generate the instruction prompt that guides the CEM opti-    Reviwer D3YD-W2
mization process. The template of the prompt is as follows:

```
            An example of the detailed prompt for instructing CEM
            process is as follows:

            Analyze the CEM optimization process and provide improvement
            suggestions.

            Important Background:  The current system uses a triple-task
            mixed reward for optimization, where mixed reward =
            Population prediction task R² * weight + Housing price
            prediction task R² * weight + GDP prediction task R² *
            weight.  All "rewards" and "performance" metrics refer to
            this mixed reward value.

            Current 3-round optimization summary:
            current_summary

            Global optimization history summary:
            limited_history

            Please provide the following content:
            1.  Analysis of the current triple-task mixed reward
            optimization state, particularly focusing on whether local
            optimum problems exist
            2.  Identify which POI categories significantly affect
            triple-task mixed performance (positive or negative)
            3.  Specific suggestions on how to adjust CEM parameters:
            - For categories with the greatest weight impact, suggest
            significant adjustments (±0.5 or more)
            - For categories with moderate weight impact, suggest
            moderate adjustments (±0.2 to ±0.4)
            - Whether smoothing_factor needs adjustment, considering
            more aggressive exploration strategies
            - Whether elite_fraction needs adjustment
            - Provide larger standard deviation (0.2--0.5) for specific
            categories to increase exploration
            4.  If optimization stagnates, suggest restarting
            distribution parameters for at least 3 categories

            Please provide specific parameter adjustment suggestions in
            JSON format as follows:
            {
            "category_adjustments":  [
            {"name":  "category_name", "mean_adjustment":  0.5,
            "std_adjustment":  0.3}
```

```
],
"global_adjustments": {
"smoothing_factor":  0.1,
"elite_fraction":  0.05
},
"restart_categories":  ["category1", "category2",
"category3"]
}
```

## D  ADDITIONAL CONTENTS OF EXPERIMENTS

### D.1  BASELINES

(1) **Baselines**

- BERT (Devlin et al., 2019b): BERT is a representative pre-trained language model that excels in capturing deep semantics. We use it to encode POIs and average for the region embedding.
- OpenAI (Neelakantan et al., 2022): OpenAI text-embedding-3-small provides high-quality text embeddings trained with large-scale contrastive objectives. We adopt it to encode POIs and aggregate for region embedding by average pooling.
- GraphSage (Hamilton et al., 2017): This classical graph learning algorithm samples and aggregates neighbor nodes to compute node embeddings. It is commonly used as a geospatial representation learning baseline with node feature or graph structure reconstruction objectives.
- DGI (Zhao et al., 2023): This method maximizes the mutual information between node and graph embeddings. We take its graph embedding as the region representation. It doesn't explicitly learn geospatial correlations.
- MVGRL (Hassani & Ahmadi, 2020): Inspired by DGI, this method maximizes the mutual information between the node and graph embedding from the original graph and an augmented graph constructed by graph diffusion. We use its graph embedding as the region representation. It doesn't explicitly learn geospatial correlations.
- HGI (Huang et al., 2023): Inspired by DGI, this method incorporates geospatial domain knowledge by hierarchically maximizing the mutual information between POI, region, and city representations. It proposes a novel rule-based strategy of positive and negative sampling to preserve fine-grained and holistic information simultaneously.
- CityFM (Balsebre et al., 2024): This method learns general-purpose geospatial representations from multimodal OpenStreetMap node, polyline, and polygon data. We use its node encoder to encode POI representations and average them as the region representation.

(2) **Model variants**

- SubUrban w/o RL: This is a variant of our model where we remove the proposed RL training process mentioned in Section 4.2 and use random selection instead.
- SubUrban w/o CEM: This is also a variant of our model where we remove the proposed LLM-instruct CEM optimization mentioned in Section 4.3.

### D.2  CROSS-TASK PERFORMANCE IN SHANGHAI

We also conduct the cross-task experiments in Shanghai. SubUrban still holds the superior performance of all tasks compared to all of the baseline methods shown in Table 6.

### D.3  GDP DENSITY PREDICTION IN SINGAPORE

In the absence of publicly available fine-grained GDP and house price datasets for Singapore and New York City, we additionally evaluate our model using an estimated Singapore GDP dataset derived from nighttime-light calibrated economic activity (Kummu et al., 2025). Using this dataset as ground truth, we report the GDP prediction performance of several competitive baselines in Table 7. As shown, SubUrban achieves the best overall performance, demonstrating strong cross-task generalization capability across cities and socioeconomic indicators.

Reviewer mxB9-Q3

Table 6: Population Density, House Price, and GDP Density Prediction in Shanghai

| Models | Population | | | House Price | | | GDP Density | | |
|---|---|---|---|---|---|---|---|---|---|
| | MAE↓ | RMSE↓ | $R^2$↑ | MAE↓ | RMSE↓ | $R^2$↑ | MAE↓ | RMSE↓ | $R^2$↑ |
| BERT-Avg | 9375.19 (±75.37) | 14235.93 (±203.54) | 0.47 (±0.01) | 15244.90 (±799.72) | 21849.39 (±1665.11) | 0.35 (±0.08) | 1461.75 (±70.10) | 2478.55 (±186.95) | 0.60 (±0.04) |
| OpenAI-Avg | 9816.80 (±108.09) | 14579.09 (±305.37) | 0.44 (±0.02) | 15566.60 (±446.11) | 22078.45 (±1158.55) | 0.35 (±0.03) | 1601.76 (±72.66) | 2644.08 (±196.10) | 0.55 (±0.02) |
| GraphSage | 8759.62 (±388.16) | 13682.10 (±644.79) | 0.53 (±0.02) | 15348.31 (±804.33) | 23770.38 (±3416.98) | 0.45 (±0.07) | 1454.01 (±74.77) | 2515.51 (±233.19) | 0.59 (±0.04) |
| DGI | 9315.73 (±441.15) | 14110.26 (±1157.06) | 0.47 (±0.05) | 15806.18 (±1539.58) | 23471.61 (±4101.79) | 0.36 (±0.09) | 1536.07 (±49.22) | 2551.83 (±125.26) | 0.60 (±0.03) |
| MVGRL | 9087.51 (±573.17) | 13646.88 (±1078.60) | 0.49 (±0.06) | 16290.52 (±923.21) | 24811.52 (±2817.58) | 0.36 (±0.05) | 1775.15 (±31.41) | 2904.99 (±142.52) | 0.48 (±0.04) |
| HGI | 7464.74 (±182.11) | 11642.35 (±289.60) | 0.66 (±0.02) | 15443.26 (±1043.29) | 24436.62 (±3630.39) | 0.42 (±0.08) | 1199.68 (±68.44) | 2247.50 (±126.82) | 0.67 (±0.02) |
| CityFM | 6558.20 (±108.37) | 10677.55 (±218.36) | 0.71 (±0.01) | 14160.05 (±692.90) | 21092.11 (±1529.19) | 0.43 (±0.06) | 867.13 (±52.74) | 1606.45 (±122.13) | 0.83 (±0.02) |
| **SubUrban** | **5684.80** (±356.93) | **9673.78** (±716.99) | **0.75** (±0.02) | **13801.49** (±1327.04) | **20511.39** (±4684.24) | **0.47** (±0.06) | **821.56** (±55.52) | **1507.51** (±87.78) | **0.84** (±0.02) |

Table 7: Nighttime-light calibrated GDP Density Prediction in Singapore

| Baseline | MAE (mean ± std) | RMSE (mean ± std) | $R^2$ (mean ± std) |
|---|---|---|---|
| BERT | 565.09 ± 17.67 | 918.74 ± 107.89 | 0.21 ± 0.06 |
| OpenAI | 576.35 ± 15.51 | 909.58 ± 104.75 | 0.22 ± 0.04 |
| CityFM | 561.99 ± 19.78 | 890.61 ± 116.28 | 0.26 ± 0.05 |
| **SubUrban** | **559.99 ± 22.63** | **836.35 ± 159.53** | **0.27 ± 0.04** |

## D.4 ANALYSIS OF REWARD SIGNAL BALANCE DURING RL TRAINING

To verify that our designs of multiple rewards from Section 4.2.3 remain balanced during optimization, we record the individual reward components across training rounds in Beijing. The values include the buffer controller reward $R_{\text{buf}}$, the multi-head attention reward $R_{\text{MHA}}$, and the GAT/Projection reward $R_{\text{GAT}}$ or $R_{\text{proj}}$. These results provide a direct view of how each module's reward evolves under our adaptive normalization and module-specific EMA baselines.

Reviewer mxB9-Q1

Table 8: Reward components across RL training rounds in Beijing

| Reward Type | Round 1 | Round 2 | Round 3 | Rdound 4 | Round 5 | Round 6 | Round 7 | Round 8 | Round 9 |
|---|---|---|---|---|---|---|---|---|---|
| $R_{\text{buf}}$ | 45.0 | 7.0 | 1.0 | 0.3 | 0.6 | 0.8 | 0.2 | 0.7 | 0.1 |
| $R_{\text{MHA}}$ | 35.0 | -4.5 | -0.2 | 0.0 | 0.4 | 0.3 | 0.0 | 0.5 | 0.0 |
| $R_{\text{GAT}}$ / $R_{\text{proj}}$ | 0.288 | 0.250 | 0.237 | 0.246 | 0.255 | 0.280 | 0.310 | 0.335 | 0.347 |

Across rounds, all three reward components rapidly converge to a consistent magnitude and evolve smoothly, demonstrating that adaptive normalization and module-specific baselines successfully stabilize the relative influence of each reward term throughout training.

## D.5 COMPARISON WITH MULTIMODAL URL BASELINE

We further compare SubUrban with multimodal urban representation learning baselines. We report the results of UrbanCLIP (Yan et al., 2024) as a representative multimodal method in Table 9. The preliminary comparison shows that SubUrban already achieves clearly superior performance, suggesting that our approach remains competitive even against multimodal models.

Reviewer D3YD-W3
Reviewer mxB9-W1

## D.6 COMPARISON WITH UNIFIED 768-D EMBEDDINGS

Reviewer oxjG-Q1&W1

Table 9: Population Density, House Price, and GDP Density Prediction in Beijing

| Models | Population | | | House Price | | | GDP Density | | |
|---|---|---|---|---|---|---|---|---|---|
| | MAE↓ | RMSE↓ | $R^2$↑ | MAE↓ | RMSE↓ | $R^2$↑ | MAE↓ | RMSE↓ | $R^2$↑ |
| UrbanCLIP | 5691.76 (±287.00) | 8571.37 (±453.97) | 0.42 (±0.07) | 21714.80 (±1717.80) | 30545.68 (±2091.35) | 0.44 (±0.10) | 949.04 (±59.51) | 1329.87 (±90.22) | -0.09 (±0.11) |
| **SubUrban** | **3283.11** (±273.61) | **5719.89** (±640.22) | **0.72** (±0.06) | **12235.97** (±1249.12) | **17021.29** (±2364.06) | **0.85** (±0.03) | **349.63** (±27.50) | **568.85** (±42.24) | **0.80** (±0.03) |

To obtain a dimensional-fair comparison result, we make an experiment that unifies all of the embedding generated from several easy-to-modify baselines to 768-D. We illustrate the results of BERT (Devlin et al., 2019a), GraphSAGE (Hamilton et al., 2017), MVGRL (Hassani & Ahmadi, 2020), CityFM (Balsebre et al., 2024), and our SubUrban in Beijing with three downstream tasks in Table 10.

Table 10: Population Density, House Price, and GDP Density Prediction in Beijing (768-D)

| Models | Population | | | House Price | | | GDP Density | | |
|---|---|---|---|---|---|---|---|---|---|
| | MAE↓ | RMSE↓ | $R^2$↑ | MAE↓ | RMSE↓ | $R^2$↑ | MAE↓ | RMSE↓ | $R^2$↑ |
| BERT-Avg (768-D) | 5043.73 (±170.77) | 8203.42 (±198.00) | 0.49 (±0.02) | 14391.39 (±861.11) | 20622.46 (±785.23) | 0.74 (±0.03) | 490.47 (±36.14) | 789.56 (±62.61) | 0.62 (±0.04) |
| GraphSAGE (768-D) | 4758.95 (±398.92) | 7516.54 (±571.75) | 0.56 (±0.05) | 14748.74 (±2750.97) | 22275.26 (±5175.66) | 0.69 (±0.17) | 488.91 (±36.89) | 782.01 (±89.28) | 0.63 (±0.04) |
| MVGRL (768-D) | 4997.34 (±221.57) | 8229.10 (±788.06) | 0.46 (±0.10) | 14048.99 (±2116.63) | 20751.86 (±4225.32) | 0.74 (±0.10) | 475.85 (±43.41) | 797.45 (±89.27) | 0.62 (±0.03) |
| CityFM (768-D) | 3912.30 (±288.16) | 6386.06 (±563.15) | 0.68 (±0.05) | 13919.86 (±1882.18) | 19523.72 (±3409.07) | 0.76 (±0.10) | 377.30 (±14.03) | 587.73 (±35.22) | 0.78 (±0.04) |
| **SubUrban (768-D)** | **3283.11** (±273.61) | **5719.89** (±640.22) | **0.72** (±0.06) | **12235.97** (±1249.12) | **17021.29** (±2364.06) | **0.85** (±0.03) | **349.63** (±27.50) | **568.85** (±42.24) | **0.80** (±0.03) |

## D.7 COMPARISON WITH UNIFIED REGION PARTITIONS

We take an experiment that unifies the region partitions by using the 3kmx3km grids to evaluate all of the methods. The results are shown in Table 11:

Reviewer t4iD-W3

Table 11: GDP Density Prediction in Beijing and Shanghai (3km x 3km Grid Region)

| Models | Beijing | | | Shanghai | | |
|---|---|---|---|---|---|---|
| | MAE↓ | RMSE↓ | $R^2$↑ | MAE↓ | RMSE↓ | $R^2$↑ |
| BERT-Avg | 110.63 (±15.30) | 261.26 (±26.68) | 0.78 (±0.07) | 310.50 (±43.63) | 799.58 (±175.18) | 0.55 (±0.15) |
| OpenAI-Avg | 129.01 (±17.24) | 263.19 (±33.77) | 0.78 (±0.04) | 355.91 (±27.48) | 850.03 (±120.69) | 0.50 (±0.07) |
| GraphSage | 100.61 (±14.33) | 240.43 (±29.49) | 0.82 (±0.04) | 313.92 (±19.13) | 847.34 (±108.76) | 0.49 (±0.12) |
| DGI | 111.76 (±7.02) | 251.53 (±30.05) | 0.82 (±0.07) | 337.87 (±115.49) | 858.13 (±277.44) | 0.62 (±0.10) |
| MVGRL | 99.69 (±6.32) | 247.43 (±15.10) | 0.83 (±0.06) | 314.96 (±107.43) | 836.19 (±262.75) | 0.65 (±0.05) |
| HGI | 103.84 (±14.33) | 235.24 (±30.82) | 0.82 (±0.03) | 244.05 (±53.15) | 596.71 (±184.52) | 0.75 (±0.06) |
| CityFM | 125.91 (±17.80) | 253.37 (±32.74) | 0.80 (±0.03) | 339.19 (±35.28) | 796.60 (±127.50) | 0.56 (±0.09) |
| **SubUrban** | **93.19** (±6.36) | **216.96** (±18.65) | **0.86** (±0.02) | **246.34** (±38.73) | **620.88** (±112.38) | **0.79** (±0.06) |

## D.8 Analysis of Marginal Gain

To provide empirical evidence supporting the submodular behavior of our reward design, we analyze the marginal gain of the mixed reward across the ten expansion rounds during the testing phase of SubUrban in Beijing. Although a formal proof of submodularity is difficult due to the heterogeneous nature of reward components, submodular functions are characterized by diminishing marginal improvements as the selection process continues. Therefore, examining the reward increments offers an intuitive evaluation of whether our system behaves in a submodular manner. Table 12 reports the mixed reward at each round and its corresponding marginal gain.

Reviewer oxjG-W4&Q4

Table 12: Mixed reward and marginal gain across expansion rounds in Beijing

| Round | Mixed Reward $R_t$ | Marginal Gain $\Delta R_t = R_t - R_{t-1}$ |
|---|---|---|
| 0 | 0.6587 | — |
| 1 | 0.7256 | +0.0669 |
| 2 | 0.7407 | +0.0151 |
| 3 | 0.7511 | +0.0104 |
| 4 | 0.7567 | +0.0056 |
| 5 | 0.7577 | +0.0010 |
| 6 | 0.7658 | +0.0081 |
| 7 | 0.7859 | +0.0201 |
| 8 | 0.7918 | +0.0059 |
| 9 | 0.7956 | +0.0038 |
| 10 | 0.7950 | -0.0006 |

The results reveal that the marginal gains decrease sharply after the first round and remain close to zero in later iterations. This consistent pattern of diminishing returns demonstrates that the optimization indeed exhibits submodular-like behavior in practice, supporting the design of our mixed reward and expansion policy.

## D.9 Ablation Study of Evaluation Models

We take an experiment that switches the evaluation model from Random Forest to MLP and Linear Regression. We take the results of Beijing with three tasks as an example. The results are shown in Table 13, which we compare all of the baselines with original dimension for generated embeddings.

Reviewer D3YD-W3
Reviewer oxjG-W2&Q2

SubUrban achieves the best performance on all tasks with both RF and MLP predictors. However, while most of the baselines perform worse with LR (e.g., MVGRL (Hassani & Ahmadi, 2020) exhibits numerical instability when fitted with LR) since these urban socioeconomic regression tasks involve strong nonlinear dependencies. Meanwhile, some baselines do not perform stably with the MLP predictor. In this case, we take the results of RF into our paper since all of the baselines perform well and are stable with this predictor.

## D.10 Ablation Study on LLM

### D.10.1 LLM Calls and Costs

We report the usage statistics and computational costs of the LLM components in SubUrban, covering two stages: (1) POI preprocessing with regional keyword generation by LLM, and (2) the CEM optimization process with LLM instructions. These results provide a transparent view of the additional overhead introduced by LLM modules in both stages.

Reviewer oxjG-W3&Q3

**LLM usage in POI preprocessing**    The total number of LLM calls in this stage equals the number of retrieved administrative regions (e.g., 16 in Beijing, 55 in Singapore). Since GPT-4 is used for generating regional keywords, we report the estimated API cost for all four cities in Table 14.

**LLM usage in the CEM process**    We further summarize the LLM calls, runtime, and estimated cost during the LLM-instructed CEM optimization stage. Results are averaged over five repeated

Reviewer oxjG-W3&Q3
Reviewer t4iD-W1

Table 13: Population Density, Houce Price, and GDP Density Prediction with Different Evaluation Models in Beijing

| Predictor | Baseline | Population | | | House Price | | | GDP Density | | |
|---|---|---|---|---|---|---|---|---|---|---|
| | | MAE↓ | RMSE↓ | R²↑ | MAE↓ | RMSE↓ | R²↑ | MAE↓ | RMSE↓ | R²↑ |
| LR | BERT | 14671.41 (±1103.29) | 23751.66 (±2214.03) | <-1 | 20279.71 (±1029.25) | 26670.96 (±1395.45) | 0.57 (±0.10) | 1250.43 (±115.20) | 1927.09 (±261.24) | <-1 |
| | OpenAI | 16735.80 (±500.19) | 26335.29 (±1684.26) | <-1 | 17404.57 (±2523.96) | 25303.11 (±4010.50) | 0.60 (±0.15) | 1333.68 (±138.98) | 2029.85 (±243.74) | <-1 |
| | DGI | 14663.68 (±4112.13) | 70117.68 (±43135.13) | <-1 | 39151.84 (±4677.73) | 55450.12 (±7601.91) | <-1 | 1185.24 (±403.94) | 4072.47 (±3753.62) | <-1 |
| | MVGRL | – | – | – | – | – | – | – | – | – |
| | GraphSage | 55415.32 (±10033.88) | 79480.10 (±13509.49) | <-1 | 18871.31 (±1051.75) | 25135.06 (±1800.14) | 0.62 (±0.07) | 4702.95 (±846.42) | 6648.30 (±1310.61) | <-1 |
| | HGI | 6244.53 (±526.02) | 9010.29 (±730.46) | 0.33 (±0.08) | 46755.54 (±6364.29) | 22185.81 (±8300.10) | <-1 | 603.31 (±65.15) | 908.26 (±100.70) | 0.51 (±0.03) |
| | CityFM | 43973.64 (±47850.59) | $3.86\times10^5$ (±$5.99\times10^5$) | <-1 | $5.23\times10^6$ (±$5.95\times10^6$) | $3.95\times10^7$ (±$4.76\times10^7$) | <-1 | 3654.18 (±3772.27) | $3.66\times10^4$ (±$5.97\times10^4$) | <-1 |
| | SubUrban | 9956.67 (±484.93) | 14202.60 (±921.89) | -0.70 (±0.13) | 22208.87 (±2895.60) | 32012.61 (±5354.17) | 0.45 (±0.25) | 922.45 (±18.66) | 1328.78 (±103.25) | -0.08 (±0.12) |
| MLP | BERT | 4462.83 (±508.57) | 7849.68 (±862.20) | 0.52 | 25106.37 (±6463.11) | 35137.16 (±8648.95) | 0.25 (±0.25) | 430.23 (±41.37) | 756.61 (±64.99) | 0.65 (±0.03) |
| | OpenAI | 4468.77 (±417.11) | 7776.44 (±725.40) | 0.52 | 23772.01 (±5234.07) | 32533.33 (±6992.70) | 0.37 (±0.20) | 423.81 (±42.08) | 756.44 (±75.70) | 0.65 (±0.03) |
| | DGI | 4599.54 (±338.89) | 8109.06 (±701.71) | 0.47 | 29267.81 (±3389.08) | 40646.66 (±4472.34) | 0.03 (±0.04) | 421.50 (±46.07) | 735.32 (±93.77) | 0.67 (±0.05) |
| | MVGRL | 5507.85 (±470.24) | 9588.58 (±1110.73) | 0.27 | 28000.79 (±4364.44) | 37813.38 (±6647.43) | 0.14 (±0.28) | 521.95 (±48.78) | 951.13 (±122.32) | 0.45 (±0.12) |
| | GraphSage | 4185.67 (±439.94) | 7528.54 (±706.94) | 0.56 | 14535.06 (±1399.61) | 19197.00 (±2353.53) | 0.78 (±0.06) | 425.49 (±47.72) | 726.45 (±78.51) | 0.68 (±0.03) |
| | HGI | 6157.84 (±532.37) | 9033.31 (±749.46) | 0.33 | 38241.42 (±6960.36) | 40792.21 (±7363.48) | <-1 | 571.67 (±77.38) | 897.86 (±112.93) | 0.52 (±0.03) |
| | CityFM | 3943.63 (±234.21) | 6942.52 (±386.02) | 0.62 | 18618.74 (±2933.79) | 24006.81 (±3688.49) | 0.66 (±0.08) | 336.48 (±23.10) | 560.64 (±43.04) | 0.81 (±0.02) |
| | SubUrban | 3793.28 (±284.90) | 6455.61 (±914.77) | 0.67 (±0.05) | 14122.74 (±1356.46) | 18126.14 (±2382.20) | 0.79 (±0.06) | **331.82 (±15.92)** | **546.52 (±41.90)** | **0.83 (±0.01)** |
| RF | BERT | 5043.73 (±170.77) | 8203.42 (±198.00) | 0.49 | 14391.39 (±681.11) | 20622.46 (±785.23) | 0.74 (±0.03) | 490.47 (±36.14) | 789.56 (±62.61) | 0.62 (±0.04) |
| | OpenAI | 5419.69 (±158.87) | 8440.61 (±172.59) | 0.46 | 13946.38 (±695.83) | 20105.17 (±1155.70) | 0.75 (±0.03) | 523.66 (±42.84) | 815.47 (±67.57) | 0.59 (±0.03) |
| | DGI | 4990.86 (±150.99) | 8153.15 (±522.52) | 0.47 | 15357.90 (±1876.32) | 20122.38 (±3558.96) | 0.75 (±0.06) | 466.77 (±23.28) | 743.05 (±74.46) | 0.67 (±0.05) |
| | MVGRL | 4990.86 (±150.99) | 8153.15 (±522.52) | 0.47 | 15692.40 (±1534.83) | 22317.73 (±3920.51) | 0.70 (±0.04) | 502.90 (±25.72) | 840.42 (±69.60) | 0.57 (±0.07) |
| | GraphSage | 4774.99 (±269.06) | 7812.94 (±578.01) | 0.52 | 14748.74 (±2750.97) | 22275.26 (±5175.66) | 0.69 (±0.17) | 488.91 (±36.89) | 782.01 (±89.28) | 0.63 (±0.04) |
| | HGI | 4534.83 (±473.15) | 7446.83 (±746.63) | 0.56 | 14719.13 (±1378.46) | 19008.63 (±1834.69) | 0.78 (±0.05) | 409.07 (±34.54) | 695.99 (±69.44) | 0.70 (±0.02) |
| | CityFM | 4199.19 (±65.02) | 6858.44 (±143.30) | 0.64 | 14291.54 (±371.40) | 19483.32 (±582.32) | 0.75 (±0.02) | 384.27 (±18.37) | 601.26 (±48.58) | 0.78 (±0.04) |
| | **SubUrban** | **3283.11 (±273.61)** | **5719.89 (±640.22)** | **0.72 (±0.06)** | **12235.97 (±1249.12)** | **17021.29 (±2364.06)** | **0.85 (±0.03)** | 349.63 (±27.50) | 568.85 (±42.24) | 0.80 (±0.03) |

Table 14: LLM calls and estimated costs for POI preprocessing

| City | #Regions (calls) | Estimated Cost (USD) |
|---|---|---|
| Beijing | 16 | 0.18 |
| Shanghai | 16 | 0.18 |
| Singapore | 55 | 0.61 |
| NYC | 5 | 0.06 |

runs in Beijing and reported in Table 15. Using LLM guidance substantially reduces optimization time while keeping costs low.

Table 15: LLM usage and cost during CEM optimization (averaged over 5 runs in Beijing)

| LLM Type | LLM Calls | Estimated Cost (USD) | Total Time (mins) | Avg Input Tokens / call |
|----------|-----------|----------------------|-------------------|-------------------------|
| No LLM | 0 | 0 | 373.60 | 0 |
| DeepSeek-R1 | 4 | 0.0196 | 286.70 | 3997.8 |
| GPT-3.5 | 4 | 0.0260 | 233.73 | 3980.2 |
| GPT-4 | 3 | 0.2411 | 191.40 | 3820.0 |

### D.10.2  LLM Types

To examine how different LLM types influence the CEM optimization process in our framework, we evaluate four settings: no LLM, DeepSeek-R1, GPT-3.5, and GPT-4. For each setting, we track the initial reward, the final reward, and the iteration at which CEM converges. All results are averaged over five runs in Beijing and shown in Table 16.

Reviewer D3YD-W3
Reviewer oxjG-W3&Q3

Table 16: CEM optimization results under different LLM types in Beijing

| LLM Type | Initial Reward | Final Reward | End Iteration |
|----------|----------------|--------------|---------------|
| No LLM | 0.4586 | 0.5272 | 13 |
| DeepSeek-R1 | 0.4855 | 0.5490 | 9 |
| GPT-3.5 | 0.4770 | 0.5796 | 11 |
| GPT-4 | 0.5033 | 0.5532 | 8 |

Overall, GPT-4 yields the strongest optimization performance with the fastest convergence, while GPT-3.5 and DeepSeek-R1 also provide notable improvements compared to using no LLM.

### D.10.3  LLM Reproducibility

To evaluate the reproducibility of the LLM-generated region keywords in the POI preprocessing stage, we conducted a stability analysis in which the same prompt template was applied five times for each city. For every pair of runs, we computed the Jaccard similarity between the generated keyword sets, where the Jaccard index measures the overlap between two sets as the size of their intersection over the size of their union. The average and standard deviation of Jaccard similarity across all five runs for each city are reported in Table 17.

Reviewer t4iD-W1

Table 17: Jaccard similarity of LLM generated regional keywords across five runs in all cities

| City | Avg Jaccard Similarity ↑ | Std |
|------|--------------------------|-----|
| Beijing | 0.83 | 0.05 |
| Shanghai | 0.84 | 0.06 |
| Singapore | 0.74 | 0.04 |
| NYC | 0.87 | 0.04 |
| **Overall** | **0.82** | **0.05** |

### D.10.4  LLM Influence

LLM intervents in two parts of SubUrban. The first part is preprocessing POI for cold-starting candidate subsets mentioned in Section 4.1, and the second part is instructing CEM optimization for attention weights of different POI categories mentioned in Section 4.3. The ablation studies of these LLM parts are based on two experiments:

**LLM Influence on regional keywords generation**  We compare how different preprocessing strategies influence the RL training dynamics. Three types of POI subsets as inputs to the hypernode expansion policy: (1) Randomly sampled subsets, (2) Subsets selected by Information Gain (Quinlan, 1986), and (3) subsets preprocessed by LLM mentioned in 4.1. The mixed rewards across training rounds in Beijing and Shanghai are reported in Table 18.

Reviewer mxB9-W2

Table 18: Mixed Rewards during RL Training under Different Preprocessing Strategies in Beijing and Shanghai

| Training Round | Beijing | | | Shanghai | | |
|:---:|:---:|:---:|:---:|:---:|:---:|:---:|
| | Random | InfoGain | LLM | Random | InfoGain | LLM |
| 1 | 0.18 | 0.22 | 0.05 | 0.42 | 0.47 | 0.45 |
| 2 | 0.34 | 0.46 | 0.20 | 0.46 | 0.44 | 0.48 |
| 3 | 0.32 | 0.30 | 0.24 | 0.51 | 0.48 | 0.55 |
| 4 | 0.36 | 0.33 | 0.27 | 0.53 | 0.54 | 0.57 |
| 5 | 0.36 | 0.32 | 0.32 | 0.54 | 0.55 | 0.58 |
| 6 | 0.36 | 0.35 | 0.36 | 0.55 | 0.52 | 0.60 |
| 7 | 0.35 | 0.35 | 0.37 | 0.54 | 0.56 | 0.61 |
| 8 | 0.37 | 0.40 | 0.38 | 0.55 | 0.58 | 0.62 |
| 9 | 0.39 | 0.41 | 0.40 | 0.57 | 0.58 | 0.63 |
| 10 | 0.41 | 0.40 | 0.44 | 0.59 | 0.60 | 0.65 |

Overall, the results show that the LLM-based preprocessing consistently yields higher rewards in later training rounds, indicating faster improvements as the RL policy evolves. These results suggest that the LLM provides a more semantically coherent and globally informed initialization, which becomes increasingly beneficial as training progresses. At the same time, we observed the strong initial performance of IG indicates that hybrid strategies (e.g., IG augmented LLM prompts) could be a promising direction for our future work.

**LLM Influence on CEM Optimization**  To quantify how different LLMs influence the CEM optimization process, we compare the reward improvements injected at each LLM-instructed iteration. The LLM is first applied at iteration 3 and then once every two iterations. Table 19 summarizes the average reward changes across these CEM iterations for four LLM settings. The results show that LLM-guided adjustments yield larger reward gains compared with the no-LLM setting, indicating that LLM feedback provides more effective directional guidance for the optimization trajectory.

Reviewer oxjG-W3&Q3

Table 19: Average reward improvement per LLM-instructed iteration during the CEM process

| LLM Type | $\Delta$(iter3→4) | $\Delta$(iter5→6) | $\Delta$(iter7→8) | $\Delta$(iter9→10) |
|:---|:---:|:---:|:---:|:---:|
| No LLM | 0.0020 | 0.0047 | 0.0067 | 0.0054 |
| DeepSeek-R1 | 0.0131 | 0.0074 | 0.0000 | 0.0000 |
| GPT-3.5 | 0.0116 | 0.0069 | 0.0124 | 0.0121 |
| GPT-4 | 0.0186 | 0.0131 | 0.0065 | 0.0252 |

Overall, these results confirm that LLM guidance improves CEM optimization effectiveness across multiple update steps.

### D.11 PARAMETER SENSITIVITY ANALYSIS

We evaluate the parameter sensitivity of SubUrban on two hyperparameters, which are the penalty coefficient $\alpha$ and Top-K in each round of expansion for each region. The penalty coefficient $\alpha$ in the Buffer Controller (Eq. 9) controls how strongly buffer expansion is penalized during RL training, while the Top-K parameter in the two-stage policy network (Section 4.2.2) determines how many POI candidates are extended per round. The details of the sensitivity results are shown in Figure 5.

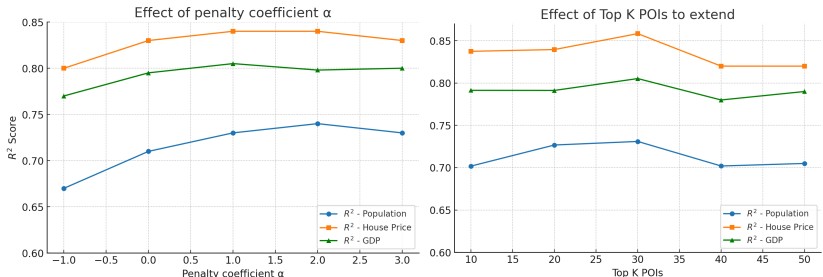

Figure 5: Parameter sensitivity analysis: (Left) Effect of penalty coefficient $\alpha$ in Buffer Controller; (Right) Effect of Top K POIs to extend on R² for population and house price prediction in Beijing.

### D.12 ANALYSIS OF A CASE REGION WITH EXPANSION

We randomly select a region (ID:111) in Beijing with a high population density as our observation target. We compare the Original Region, Random Expansion, SubUrban expansion with population task reward as feedback only (SubUrban_Pop), and SubUrban with the combined reward of triple tasks as feedback (SubUrban_Triple). The average buffer distance after 10 rounds of expansion is around 3 kilometers for each region in Beijing.

From the spatial aspect, visualizations are shown in Figure 6. Each figure illustrates the spatial distributions of POIs after 10 rounds of expansion. Different colors represent the categories of extended POIs around this region. Compared to the Random Expansion, the spatial distribution of expanded POIs are more evenly distributed in geographical space with a few clusters, which proves that the RL-trained model ensures a less biased and spatially balanced exploration space due to the coverage restriction in the definition of the state.

From the semantic aspect, statistics of POIs categories after expansion are shown in Figure 7. The grey bars in the histogram represent the original distribution of POI categories, blue bars represent the LLM preselected POI categories, while orange bars represent the expanded categories of POIs. Firstly, based on the pre-trained and retrieved knowledge for this region, LLM distinguishes that categories such as "Address", "Companies", and "Government" are especially relevant to the functionality of this region, so that it keeps these POIs more than others. Secondly, SubUrban variants further focus on a smaller set of categories compared with Random Expansion, suggesting a tendency to concentrate on task-relevant semantics rather than aimless diversification. Thirdly, SubUrban_Pop expands more "Shopping" POIs, which is intuitively consistent with the strong connection between shopping activities and population density, while SubUrban_Triple shifts toward "Public" and "CarSales" categories, reflecting additional relevance to GDP and housing price prediction.

In summary, these spatial and semantic results confirm that SubUrban does not expand POIs arbitrarily, but instead learns to autonomously balance spatial coverage, semantic focus, and task-specific relevance in a way that is both interpretable and practically meaningful.

## E DISCLOSURE OF LLM USAGE

We made limited use of GPT-5 for editing purposes, specifically to enhance clarity and grammar of the text. All core aspects of this research, including idea formulation, experimental methodology, and result interpretation, were conducted without LLM assistance.

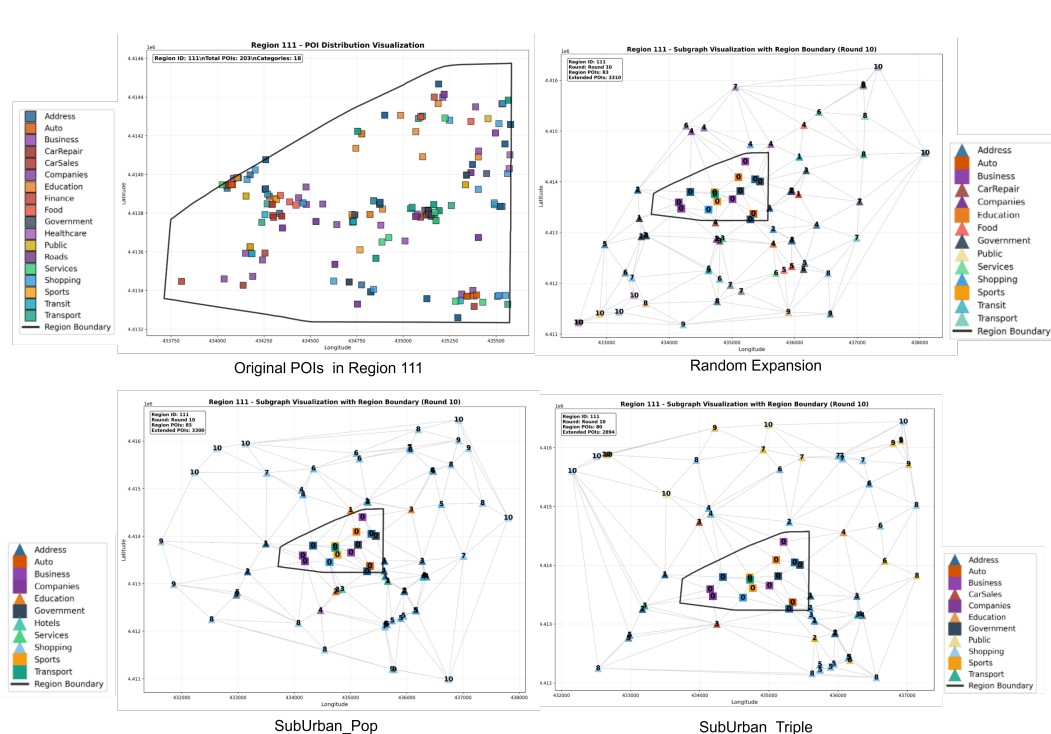

Figure 6: Visualizations of Original vs. Random Expansion vs. SubUrban_Triple vs. SubUrban_Pop.

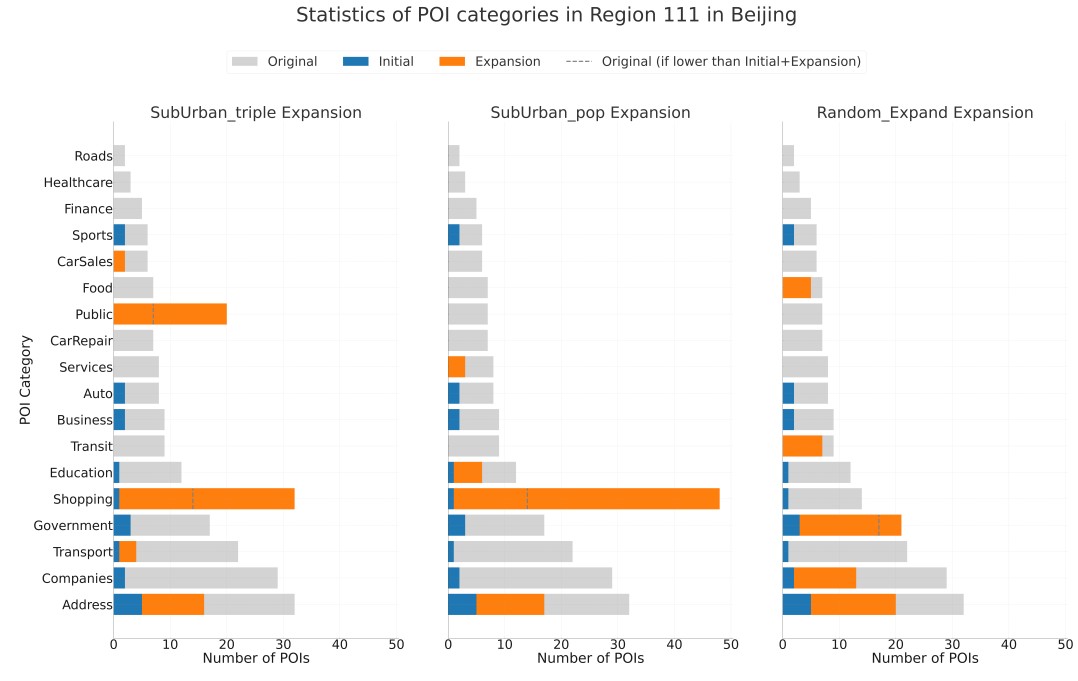

Figure 7: Statistics of expanded POI categories from SubUrban_Triple vs. SubUrban_Pop vs. Random Expansion.

