# OpenReview forum: "Autonomous Urban Region Representation with LLM-informed Reinforcement Learning"
_ICLR.cc/2026/Conference — Submitted to ICLR 2026_

### Official Review · Reviewer_t4iD · 2025-10-26

**Soundness:** 1
**Presentation:** 1
**Contribution:** 1
**Rating:** 0
**Confidence:** 5

**Summary:**

**IMPORTANT: The anonymous code repo was last updated on Oct. 1, 2025, which is a few days after the paper and supplemantary material submision deadline. I believe this violates ICLR code of conducts and should be desk rejected. The review below is only for reference.**

This paper proposes SubUrban, an RL-based framework for urban region representation learning, aiming to reduce reliance on manual feature engineering and city-specific heuristics. The propsoed approach includes 1) LLM-guided POI preprocessing to filter redundant or low-value urban features, 2) a submodular-aware hypernode expansion mechanism to adaptively construct expressive regional representations, and 3) an LLM-instructed CEM optimization strategy to calibrate category-wise attention weights. Experiments conducted on four cities (Beijing, Shanghai, Singapore, NYC) and three prediction tasks (population, house price, GDP density) demonstrate improved performance and robustness.

**Strengths:**

1. Overall, the paper is well-organized, and the motivation is clearly stated from the standpoint of reducing human-designed heuristics.
2. Experimental results are extensive, involving multiple cities and tasks, and the reported data efficiency improvements seem to be promising.

**Weaknesses:**

1. While the model claims to reduce heuristic dependency, involving a LLM naturally introduces a new form of heuristic (e.g., manually designed prompt templates, assumed semantic priors of regions). Clear clarification is needed to demonstrate how stable or reproducible these LLM-based components are across different language models or prompt variations.
2. The description of hypernode expansion (soft/hard selection alternation) is technically detailed, but the intuitions behind key parameters are relatively under-explained. It is generally more important to explain "why" ratehr than simply introducing "how". Besides, it is unclear how sensitive the performance is to these hyperparameters.
3. Although experiments are conducted on four cities, the source and partitioning methods differ (GADM vs. OSM vs. NYC planning). It could be helpful to study and clarify whether these differences influence evaluation comparability.

**Questions:**

Please refer to the weakness section for my questions.

---

> ### Author Response · Authors · 2025-11-26
>
> We sincerely thank you for taking the time to review our work. We address your comments and questions below.
>
> ### IMPORTANT: Update of anonymous code repo after paper DDL
>
> We appreciate your attention to the conference policies. You raised the concern that an update to our anonymous repository after the paper and supplementary material submission deadline violates the ICLR Code of Conduct. However, we believe there may be some misunderstandings.
>
> - First, the update in question did not modify any supplementary materials. The only change made after the deadline was adjusting the visibility of the anonymous repository.
>
> - Second, the reviewer does not cite any specific ICLR policy that prohibits updating the visibility of an external anonymous repository. After carefully re-reading the ICLR author guidelines and Code of Conduct, we could not find any rule indicating that such a change would constitute a violation or grounds for desk rejection. In contrast, the paper and supplementary materials are even allowed to be further revised during the discussion phase.
>
> - Third, while we appreciate the positive summary of our paper’s strengths, we noticed that the assigned scores for Soundness, Presentation, and Contribution were unusually low, and the overall rating is astonishingly bad. This appears to be improperly affected by the desk rejection claims, conflicts with the content of the written review, and may not accurately reflect the assessment described in the comments.
>
> We would greatly appreciate it if you could reference the specific guideline or policy that we have violated. And we expect you to score objectively and fairly. We will fully comply with all decisions made by the ICLR official committee regarding desk rejection.

---

> > ### Author Response · Authors · 2025-11-26
> >
> > ### W1. LLM types and prompt variations
> >
> > - LLM usage in keyword generation for POI preprocessing
> >
> >     To evaluate the reproducibility of the LLM-generated region keywords in the POI preprocessing stage, we conducted an additional stability analysis. For each city, we applied the same prompt template and repeated the keyword generation five times. We then computed the Jaccard similarity between the keyword sets produced across runs, where Jaccard similarity measures the overlap between two sets as the ratio of their intersection over their union. The results are shown below:
> >
> >     | City       | Avg Jaccard similarity ↑| Std      |
> >     | ---------- | ---------------------- | -------- |
> >     | Beijing    | 0.83                   | 0.05     |
> >     | Shanghai   | 0.84                   | 0.06     |
> >     | Singapore  | 0.74                   | 0.04     |
> >     | NYC        | 0.87                   | 0.04     |
> >     | **Overall** | **0.82**              | **0.05** |
> >
> >
> > - LLM usage in CEM optimization process
> >
> >     For LLM used in CEM optimization process, the detailed LLM usage statistics for the CEM stage in Beijing are shown below, reported as the average over 5 repeated runs:
> >
> >     | LLM Type    | LLM Call Count | Estimated Cost (USD)  | Total Time (min)  | Avg Input Tokens (per call) |
> >     |-------------|----------------|-----------------------|--------------------|------------------|
> >     | No LLM      | 0              | 0                     | 373.6              | 0                |
> >     | DeepSeek-R1 | 4              | 0.0196                | 286.7              | 3997.8           |
> >     | GPT-3.5     | 4              | 0.0260                | 233.73             | 3980.2           |
> >     | GPT-4       | 3              | 0.2411                | 191.4              | 3820.0
> >
> > - Summary
> >
> >     Both tables show that the LLM components in SubUrban are stable and reproducible. The LLM usage in both stages is lightweight, requires no sensitive prompt tuning, and provides reliable semantic guidance without introducing heuristics.
> >
> > ### W2. Key hyperparameters in hypernode expansion
> >
> > For the process of the hypernode expansion, we design a two-step action (soft selection + hard selection, in Section 4.2.2) to expand each region to form the hypernode. The key hyperparameters are discussed in Parameter Sensitivity Analysis in Section 5.3, which include two hyperparameters as below:
> >
> > - Penalty coefficient α
> >
> >     The penalty coefficient α in the Buffer Controller controls how strongly buffer expansion is penalized during RL training. The actual buffer value is dynamically learned by the RL agent based on task requirements and regional characteristics, with α modulating the penalty strength to balance performance and efficiency.
> >
> > - Top-K
> >
> >     The Top-K parameter in the two-stage policy network acts as an upper bound on the number of POI extensions per region. The actual number of extensions is determined dynamically by how many candidate POIs have similarity scores exceeding the average threshold to the target region, with Top-K simply setting this upper limit.
> >
> > Both α and Top-K are introduced as controllable parameters for sensitivity testing to evaluate the model's robustness under different configurations, rather than as heuristic design choices. The sensitivity analysis of these two hyperparameters is shown in Figure 7, which proves the insensitivity of SubUrban to these hyperparameters in hypernode expansion, and also proves the diminishing marginal effects of expansion on downstream performance.
> >
> > ### W3. Different partitioning methods in different cities
> >
> > To address potential concerns about region partitioning, we tested the grid partition, which is also frequently used in geospatial analysis. We take the unified 3km × 3km grid across both Beijing (1873 grid regions) and Shanghai (853 grid regions) to evaluate all of the methods with the task of GDP Density Prediction. The results are shown below:
> >
> > | Model            | Beijing R² ↑       | Shanghai R² ↑      |
> > | ---------------- | ------------------ | ------------------ |
> > | BERT-Avg         | 0.78 ± 0.07        | 0.55 ± 0.15        |
> > | OpenAI-Avg       | 0.78 ± 0.04        | 0.50 ± 0.07        |
> > | GraphSAGE        | 0.82 ± 0.04        | 0.49 ± 0.12        |
> > | DGI              | 0.82 ± 0.07        | 0.62 ± 0.10        |
> > | MVGRL            | 0.83 ± 0.06        | 0.65 ± 0.05        |
> > | HGI              | 0.83 ± 0.03        | 0.75 ± 0.06        |
> > | CityFM           | 0.80 ± 0.03        | 0.56 ± 0.09        |
> > | **SubUrban**     | **0.86 ± 0.02**    | **0.79 ± 0.06**    |
> >
> > It proves that SubUrban still achieves the best performance under region partitions, suggesting its good compatibility to region partitions. This is reasonable as the proposed RL-based expansion process is unlikely to be confined by the shape of downstream regions. The full results will be updated in the revised version of the paper.

---

### Official Review · Reviewer_oxjG · 2025-10-31

**Soundness:** 3
**Presentation:** 3
**Contribution:** 3
**Rating:** 4
**Confidence:** 3

**Summary:**

This paper presents SubUrban, a framework that combines submodular rewards with reinforcement learning and incorporates LLM-based semantic guidance in preprocessing and parameter search. The system first uses LLM-generated keywords and clustering to semantically pre-filter large POI sets, then treats the filtered POIs (by category) as actions and defines a reward balancing coverage, saturation, and buffer to train a modular policy that selects the most informative POIs under a budget to expand hypernodes. Category-weight search is accelerated with an LLM-guided Cross-Entropy Method (CEM). The authors evaluate on Beijing, Shanghai, Singapore, and New York across downstream regression tasks (population density, house price, GDP), including sparse-data settings (e.g., using only 10% of POIs), and report that SubUrban outperforms several strong baselines in many settings while offering data efficiency and interpretability; implementation details and appendices are provided and the authors commit to open-sourcing the code.

**Strengths:**

1. The idea is clear and practical: using submodularity to model diminishing marginal returns of POI selection and learning policies under budget constraints via RL is intuitive and engineering-ready.

2. The LLM-in-the-loop engineering attempt is valuable: using LLMs for semantic prefiltering and to guide CEM reduces manual heuristics and has practical appeal.

3. Broad empirical coverage: comparisons and ablations across four cities, several regression tasks, and sparse-data scenarios (e.g., 10% POIs) demonstrate applicability in varied settings.

4. Interpretability and intuitive design: the Coverage/Saturation/Buffer components help explain why certain POIs are selected, aiding qualitative analysis and visualization.

**Weaknesses:**

1. Different baselines in the paper use embeddings of varying dimensionalities (e.g., BERT 768, OpenAI 1536, HGI 64, CityFM 1024), which can significantly influence downstream Random Forest performance and lead to unfair comparisons.

2. The study relies solely on Random Forest (with a 4:1 train–test split) as the downstream evaluator, without demonstrating results from stronger or more diverse supervised learners (e.g., MLP, GBDT/XGBoost, or a linear regression baseline). This may overestimate or underestimate the embedding quality.

3. Although LLM prompts and templates are provided in Appendix C.1/C.2, the main text omits crucial operational statistics—such as the number of LLM calls, average query size, total token cost, and whether any manual filtering of outputs was performed.

4. While the paper frequently refers to “submodular gains” and “marginal utilities” to motivate its selection strategy, it does not provide a formal proof or sufficient conditions showing that the designed reward function or policy is truly submodular. If submodularity does not hold, the approximation guarantees of the greedy policy become invalid.

**Questions:**

1. The appendix indicates substantial differences in embedding dimensionality across baselines (e.g., BERT 768, OpenAI 1536, HGI 64, CityFM 1024), which could affect the fairness of downstream comparisons. Have the authors attempted to unify or project these embeddings to a common dimension? If not, please consider adding unified-dimension experiments or a sensitivity ablation, and specify this in the tables.

2. The current experiments primarily rely on Random Forest as the downstream evaluator. To provide a more comprehensive assessment of embedding quality, it would be valuable to include results from other evaluators (e.g., linear regression, MLP, XGBoost/LightGBM, or end-to-end fine-tuning) and indicate whether the main conclusions hold consistently across these setups.

3. The paper mentions using different LLMs at various stages (e.g., prefiltering and CEM optimization), yet the corresponding statistics remain somewhat abstract. It would be helpful if the authors could provide a systematic summary of the LLM models/versions used at each stage, along with call counts, average tokens, total runtime, and estimated cost. Including ablation results such as no-LLM / small-LLM / GPT-4 in the appendix would further clarify the performance–cost trade-off introduced by LLM integration.

4. The discussion of submodularity in the reward function is mostly intuitive and lacks explicit theoretical assumptions or validation. Under what conditions can the reward be guaranteed to be submodular? If a formal proof is challenging, please consider providing marginal-gain curves or statistics for representative regions to demonstrate approximate submodularity.

---

> ### Author Response · Authors · 2025-11-26
>
> Thank you very much for your thoughtful and encouraging review. Your feedback greatly summarizes the principle and framework of our work. We really appreciate the constructive points for several ablations you raised, and our answers are as follows:
>
> ### W1 and Q1. Influence of the dimension setting of the generated embedding
>
> As you reorganized from Appendix B, the baselines inherently use different dimensions (e.g., BERT 768, OpenAI 1536, HGI 64, CityFM 1024), and we originally kept their native settings to reproduce their reported performance faithfully. While your concern about the unfair comparisons is insightful, we did a quick test of three tasks in Beijing with the unified 768-D on our SubUrban and several easy-to-modify methods. The table below reports the R² results for simplicity.
>
> | Model            | Population R² ↑ | House Price R² ↑ | GDP Density R² ↑ |
> |------------------|---------------|----------------|----------------|
> | BERT-Avg         | 0.49 ± 0.02   | 0.74 ± 0.03    | 0.62 ± 0.04    |
> | GraphSAGE        | 0.56 ± 0.05   | 0.69 ± 0.17    | 0.63 ± 0.04    |
> | MVGRL            | 0.46 ± 0.10   | 0.74 ± 0.10    | 0.62 ± 0.03    |
> | CityFM           | 0.68 ± 0.05   | 0.76 ± 0.10    | 0.78 ± 0.04    |
> | **SubUrban**     | **0.72 ± 0.06** | **0.85 ± 0.03** | **0.80 ± 0.03** |
>
> The results reflect the consistent best performance of our SubUrban when the dimension of embedding is unified.
>
> The full results will be added to the updated PDF of our paper.

---

> > ### Author Response · Authors · 2025-11-26
> >
> > ### W2 and Q2. Influence of the different evaluators
> >
> > To further prove the embedding quality, we add the experiments using Multilayer Perceptron (MLP) and Linear Regression (LR) as predictors, and the comparative results on three tasks in **Beijing** using different predictors are shown below:
> >
> > | Predictor | Baseline  | Pop-MAE     | Pop-R²   | House-MAE    | House-R²    | GDP-MAE    | GDP-R²   |
> > | --------- | --------- | ----------- | -------- | ------------ | ----------- | ---------- | -------- |
> > | LR        | BERT      | 14671.41    | < -1   | 20279.71     | 0.57        | 1250.43    | < -1    |
> > | LR        | OpenAI    | 16735.80    | < -1   | 17404.57     | 0.60        | 1333.68    | < -1    |
> > | LR        | DGI       | 14663.68    | < -1   | 39151.84     | -0.86       | 1185.24    | < -1    |
> > | LR        | MVGRL     | –           | –        | –            | –           | –          | –        |
> > | LR        | GraphSage | 55415.32    | < -1   | 16871.31 | 0.62    | 4702.95    | < -1  |
> > | LR        | HGI       | 6244.53 | 0.33 | 46755.54     | < -1       | 603.31 | 0.51 |
> > | LR        | CityFM    | 43973.64    | < -1 | > 10^6  | < -1 | 3654.18    | < -1 |
> > | LR        | SubUrban  | 9956.67     | -0.70    | 22208.87     |   0.45        | 922.45     | -0.08    |
> > | --------- | --------- | ----------- | -------- | ------------ | -------- | ---------- | -------- |
> > | MLP       | BERT      | 4462.83     | 0.52     | 25106.37     | 0.25     | 430.23     | 0.65     |
> > | MLP       | OpenAI    | 4468.77     | 0.52     | 23772.01     | 0.37     | 423.81     | 0.65     |
> > | MLP       | DGI       | 4599.54     | 0.47     | 29267.81     | 0.03     | 421.50     | 0.67     |
> > | MLP       | MVGRL     | 5507.85     | 0.27     | 28000.79     | 0.14     | 521.95     | 0.45     |
> > | MLP       | GraphSage | 4185.67     | 0.56     | 14535.06     | 0.78     | 425.49     | 0.68     |
> > | MLP       | HGI       | 6157.84     | 0.33     | 38241.42     | < -1     | 571.67     | 0.52     |
> > | MLP       | CityFM    | 3943.63     | 0.62     | 18618.74     | 0.66     |   336.48   | 0.81     |
> > | MLP       | SubUrban  | 3793.28 | 0.67 | 14122.74 | 0.79 | **331.82** | **0.83** |
> > | --------- | --------- | ------- | ------ | --------- | -------- | ------- | ------ |
> > | RF        | BERT      | 5043.73 | 0.49   | 14391.39  | 0.74     | 490.47  | 0.62   |
> > | RF        | OpenAI    | 5419.69 | 0.46   | 13946.38  | 0.75     | 523.66  | 0.59   |
> > | RF        | DGI       | 4990.86 | 0.47   | 15357.90  | 0.75     | 466.77  | 0.67   |
> > | RF        | MVGRL     | 4990.86 | 0.47   | 15692.40  | 0.70     | 502.90  | 0.57   |
> > | RF        | GraphSage | 4774.99 | 0.52   | 14748.74  | 0.69     | 488.91  | 0.63   |
> > | RF        | HGI       | 4534.83 | 0.56   | 14719.13  | 0.78     | 409.07  | 0.70   |
> > | RF        | CityFM    | 4199.19 | 0.64   | 14291.54  | 0.75     | 384.27  | 0.78   |
> > | RF        | **SubUrban**  | **3283.11** | **0.72**   | **12235.97**  | **0.85**     | 349.63  | 0.80   |
> >
> > **SubUrban shows the most stable and consistent performance across all evaluators**. It achieves the best performance under the stable RF evaluator and complex MLP evaluator. For Linear Regression, most of the baselines do not perform well (e.g., MVGRL exhibits numerical instability when fitted with LR) because LR can only capture linear relationships, while these urban socioeconomic prediction tasks involve strong nonlinear spatial dependencies. Meanwhile, some baselines do not perform stably with the MLP predictor. In this case, we take the results of RF into our paper since all of the baselines perform well and are stable with this predictor.
> >
> > For other evaluators such as GBDT and XGBoost, exploring more sophisticated predictors goes beyond the main scope of our work. Nevertheless, we appreciate the reviewer’s suggestion, and this could be a valuable direction for future research. The full results will be added to the updated PDF of our paper.

---

> > > ### Author Response · Authors · 2025-11-26
> > >
> > > ### W3 and Q3. Statistics and Ablation of LLM Usage
> > >
> > > - LLM usage and cost during POI Preprocessing
> > >
> > >     The number of calls in the POI preprocessing stage is determined by the number of retrieved regions (e.g., 16 administrative regions in Beijing and 5 boroughs of NYC). As we use GPT-4 for regional keywords generation, we list the statistics of LLM usage and cost of POI preprocessing on four cities as below:
> > >
> > >     | City       | #Regions (calls) | Estimated Cost (USD) |
> > >     |------------|------------------|-----------------------|
> > >     | Beijing    | 16               | 0.18                  |
> > >     | Shanghai   | 16               | 0.18                  |
> > >     | Singapore  | 55               | 0.61                  |
> > >     | NYC        | 5                | 0.06                  |
> > >
> > > - LLM usage and cost during CEM process
> > >
> > >     The detailed LLM usage statistics for the CEM stage in Beijing are summarized below, reported as the average over 5 repeated runs:
> > >
> > >     | LLM Type    | LLM Call Count | Estimated Cost (USD)  | Total Time (mins)  | Avg Input Tokens (per call) |
> > >     |-------------|----------------|-----------------------|--------------------|------------------|
> > >     | No LLM      | 0              | 0                     | 373.6              | 0                |
> > >     | DeepSeek-R1 | 4              | 0.0196                | 286.7              | 3997.8           |
> > >     | GPT-3.5     | 4              | 0.0260                | 233.73             | 3980.2           |
> > >     | GPT-4       | 3              | 0.2411                | 191.4              | 3820.0           |
> > >
> > >     We also report the statistics of the mixed rewards during the LLM-instructed CEM process in Beijing. The table below summarizes the average initial reward, final reward, and the iteration at which the process converges.
> > >
> > >
> > >     | LLM Type | Initial Reward (avg) | Final Reward (avg) | End Iteration |
> > >     |-------------|-----------------------|---------------------|----|
> > >     | No LLM      | 0.4586                | 0.5272              | 13 |
> > >     | DeepSeek-R1 | 0.4855                | 0.5490              | 9  |
> > >     | GPT-3.5     | 0.4770                | 0.5796              | 11 |
> > >     | GPT-4       | 0.5033                | 0.5532              | 8  |
> > >
> > >     The following table summarizes the average reward changes at each LLM instructed iteration during the CEM process, with the LLM first applied at iteration 3 and then once every two iterations.
> > >
> > >     | LLM Type    | Δ(iter3→4)  | Δ(iter5→6)  | Δ(iter7→8)  | Δ(iter9→10) |
> > >     |-------------|---------|---------|---------|---------|
> > >     | No LLM      | 0.0020  | 0.0047  | 0.0067  | 0.0054  |
> > >     | DeepSeek-R1 | 0.0131  | 0.0074  | 0.0000  | 0.0000  |
> > >     | GPT-3.5     | 0.0116  | 0.0069  | 0.0124  | 0.0121  |
> > >     | GPT-4       | 0.0186  | 0.0131  | 0.0065  | 0.0252  |
> > >
> > > - Summary
> > >
> > >     Overall, LLM usage in both the POI preprocessing and CEM stages incurs only minimal cost. POI preprocessing requires just one short GPT-4 call per region, resulting in negligible expense across all cities. In the CEM stage, GPT-4 achieves the fastest convergence and greatest improvements at a slightly higher cost, while GPT-3.5 and DeepSeek-R1 provide competitive gains with very low cost. Across all cases, LLM-instructed variants outperform the No-LLM baseline in both convergence speed and final reward.
> > >
> > >     The full results will be added to the updated PDF of our paper.
> > >
> > > ### W4 and Q4. Marginal-gain statistics
> > >
> > > A formal proof of submodularity is difficult because the reward combines several heterogeneous components. Instead, we provide empirical evidence that the optimization behaves in a submodular manner. To prove the effectiveness of rewards, we give the detailed Marginal Gain from the growth of Mixed Reward during the testing phase of SubUrban in Beijing.
> > >
> > > | Round | Mixed Reward Rₜ | ΔRₜ = Rₜ − Rₜ₋₁ |
> > > |-------|------------------|------------------|
> > > | 0     | 0.6587           | –                |
> > > | 1     | 0.7256           | +0.0669          |
> > > | 2     | 0.7407           | +0.0151          |
> > > | 3     | 0.7511           | +0.0104          |
> > > | 4     | 0.7567           | +0.0056          |
> > > | 5     | 0.7577           | +0.0010          |
> > > | 6     | 0.7658           | +0.0081          |
> > > | 7     | 0.7859           | +0.0201          |
> > > | 8     | 0.7918           | +0.0059          |
> > > | 9     | 0.7956           | +0.0038          |
> > > | 10    | 0.7950           | −0.0006          |
> > >
> > > The results show that the marginal gains of Mixed Reward rapidly decrease from round 1 to values close to zero in later rounds. This clear pattern of diminishing marginal gains indicates a submodular manner in practice.

---

### Official Review · Reviewer_mxB9 · 2025-10-31

**Soundness:** 3
**Presentation:** 2
**Contribution:** 3
**Rating:** 4
**Confidence:** 3

**Summary:**

This paper proposes a self-supervised learning paradigm based on submodular functions and reinforcement learning, which models POI selection as a sequential decision-making process. By defining states such as Coverage, Saturation, and Buffer, and incorporating reward signals that combine downstream task performance with improvements in local states, it autonomously learns an expansion strategy, thereby reducing reliance on manual feature engineering and heuristic design. It introduces LLMs to provide semantic guidance in the urban domain, including generating representative keywords during the preprocessing stage to filter the initial POI candidate set, and guiding the Cross-Entropy Method during the optimization stage to adjust the attention weights for POI categories, consequently accelerating convergence and enhancing cross-city transferability. Experiments across multiple cities and downstream tasks demonstrate that SubUrban outperforms existing state-of-the-art methods using only 10% of the data, exhibiting exceptional data efficiency, robustness across cities and tasks, and interpretability.

**Strengths:**

1. This paper innovatively combines submodular functions with the sequential decision-making capability of RL for autonomous construction of urban hypernodes. This approach offers a novel and automated perspective to address the long-standing pain points in urban computing that rely on manual heuristics and city-specific tuning. The utilization of LLMs to inject domain knowledge for guiding data selection and optimization processes is also an interesting methodology.
2. The proposed framework in this paper demonstrates high practical value, as it can significantly reduce the costs associated with data processing and model tuning for urban AI applications, while enhancing the model's generalization capability across cities with varying data distributions. This is crucial for the scalable deployment of smart city applications.

**Weaknesses:**

1. The study primarily compares POI-encoding-based representation learning methods, which is reasonable given its core focus on processing POIs. However, incorporating some powerful multimodal fusion methods (such as UrbanCLIP or UrbanVLM) that also generate high-quality regional representations as baselines-or comparing/combining SubUrban's learned representations with those from such models-could yield more compelling evidence.
2. The entire system integrates multiple complex components including RL, submodular rewards, LLM preprocessing, and LLM-instructed CEM. Although ablation studies were conducted, it remains unclear, for instance, to what extent the LLM contributes. How much would performance degrade if the LLM were replaced with a simple statistics-based method (e.g., using information gain) to generate initial keywords? Such analysis would help determine whether the LLM truly provides irreplaceable semantic understanding or merely offers a decent initialization.

**Questions:**

1. The paper designs multiple reward signals. In practical training, how are these reward terms (e.g., $R_{GAT}$ , $R_{MHA}$ , $R_{buf}$ ) balanced during optimization to prevent any single component from dominating the entire training process?
2. The case study in Section D.5 is insightful. Could you briefly comment on whether the expansion strategies learned by SubUrban demonstrate consistent and interpretable patterns across the multiple regions you observed? For instance, are there systematic differences in the focused POI categories and spatial expansion patterns for different functional area types, such as residential versus commercial zones?
3. Are complete results for House Price and GDP Density prediction available for Singapore and NYC?

---

> ### Author Response · Authors · 2025-11-26
>
> Thank you for the thoughtful and encouraging review. We appreciate your recognition of the strengths of our framework and practical values for the scalable deployment of smart city applications. We now respond to the questions you raised.
>
>
> ### W1. Multimodal fusion methods
>
> We thank the reviewer for the valuable suggestions. We are currently trying to incorporate the listed methods into our pipeline. We give the results of three tasks by UrbanCLIP in **Beijing**, where we also put the performance of our SubUrban here for comparison.
>
> | Baseline   | POP-MAE | POP-RMSE | POP-R² | HOUSE-MAE | HOUSE-RMSE | HOUSE-R² | GDP-MAE | GDP-RMSE | GDP-R² |
> |------------|---------|----------|--------|-----------|------------|----------|---------|----------|--------|
> | UrbanCLIP  | 5691.76 | 8571.37  | 0.42 | 21714.80  | 30545.68   | 0.44   | 949.04  | 1329.87  | -0.09 |
> | SubUrban   | 3283.11 | 5719.89  | 0.72   | 12235.97  | 17021.29   | 0.85     | 349.63  | 568.85   | 0.80    |
>
> These results show that although UrbanCLIP leverages both remote-sensing imagery and textual descriptions, its downstream performance still does not exceed that of our POI-only SubUrban.
>
> For the results of more multimodal fusion baselines, we will update the paper once the experiments are completed.
>
> ### W2. Information gain vs. LLM for POI preprocessing
>
> Thank you for the insightful suggestion. We agree that our use of the LLM component should be evaluated more carefully. Following this suggestion, we conducted additional experiments comparing the LLM-based preprocessing with a statistics-based method: Information Gain (IG), and the random sampling baseline. The mixed rewards with using different methods processed POIs input during the RL training phase in **Beijing** and **Shanghai** are shown as follows:
>
> - Mixed Rewards during RL training phase in Beijing:
>
>     | Training Round | Random Sample | Information Gain | LLM Preprocess |
>     |----------------|---------------|------------------|----------------|
>     | 1              | 0.18          | 0.22             | 0.05           |
>     | 2              | 0.34          | 0.46             | 0.20           |
>     | 3              | 0.32          | 0.30             | 0.24           |
>     | 4              | 0.36          | 0.33             | 0.27           |
>     | 5              | 0.36          | 0.32             | 0.32           |
>     | 6              | 0.36          | 0.35             | 0.36           |
>     | 7              | 0.35          | 0.35             | 0.37           |
>     | 8              | 0.37          | 0.40             | 0.38           |
>     | 9              | 0.39          | 0.41             | 0.40           |
>     | 10             | 0.41          | 0.40             | 0.44           |
>
> - Mixed Rewards during RL training phase in Shanghai:
>
>     | Training Round | Random Sample | Information Gain | LLM Preprocess |
>     |----------------|---------------|------------------|----------------|
>     | 1              | 0.42          | 0.47             | 0.45           |
>     | 2              | 0.46          | 0.44             | 0.48           |
>     | 3              | 0.51          | 0.48             | 0.55           |
>     | 4              | 0.53          | 0.54             | 0.57           |
>     | 5              | 0.54          | 0.55             | 0.58           |
>     | 6              | 0.55          | 0.52             | 0.60           |
>     | 7              | 0.54          | 0.56             | 0.61           |
>     | 8              | 0.55          | 0.58             | 0.62           |
>     | 9              | 0.57          | 0.58             | 0.63           |
>     | 10             | 0.59          | 0.60             | 0.65           |
>
> Overall, the results show that the LLM-based preprocessing consistently yields higher rewards in later training rounds, indicating faster improvements as the RL policy evolves. These results suggest that the LLM provides a more semantically coherent and globally informed initialization, which becomes increasingly beneficial as training progresses. At the same time, we appreciate the reviewer’s insight: the strong initial performance of IG indicates that hybrid strategies (e.g., IG augmented LLM prompts) could be a promising direction for our future work.

---

> > ### Author Response · Authors · 2025-11-26
> >
> > ### Q1. Balance of Reward Signals
> >
> > Our design prevents any reward component from dominating optimization through two built-in mechanisms described in the Reward Function and Advantage Function in Section 4.2.3. The reward definitions in Equations (7) - (9) apply adaptive normalization using historical standard deviations, which keep the scales of semantic diversity, spatial coverage, downstream performance, and buffer signals comparable without manual tuning. The advantage formulation described in Equation (10) provides each module with its own adaptive EMA baseline, so optimization depends only on deviations from its expected reward rather than on raw magnitudes. We also print the detailed rewards during RL training process for Beijing as below:
> >
> > | Training Round | R_buf | R_MHA | R_GAT / R_proj |
> > | -------------- | ----- | ----- | -------------- |
> > | 1              | 45.0  | 35.0  | 0.288          |
> > | 2              | 7.0   | -4.5  | 0.250          |
> > | 3              | 1.0   | -0.2  | 0.237          |
> > | 4              | 0.3   | 0.0   | 0.246          |
> > | 5              | 0.6   | 0.4   | 0.255          |
> > | 6              | 0.8   | 0.3   | 0.280          |
> > | 7              | 0.2   | 0.0   | 0.310          |
> > | 8              | 0.7   | 0.5   | 0.335          |
> > | 9              | 0.1   | 0.0   | 0.347          |
> >
> > The reward trends further confirm that the system remains well balanced. Although the rewards of buffer controller and multihead attention show large initial magnitudes in the first round, both quickly stabilize to the same scale as the GAT and Projection Layer rewards. The balanced trajectories across rounds demonstrate that the combination of adaptive normalization and module-specific baselines effectively regulates the relative influence of the different reward terms.
> >
> > ### Q2. Spatial Patterns Revealed by SubUrban Expansion
> >
> > Thank you for your interest in the case study in Appendix D.5. As discussed in that section, we observed consistent differences in the POI categories preferred during expansion across regions with different functional characteristics. We summarize based on our observations
> >
> > 1. Population patterns
> >
> >     High-population regions tend to expand toward categories such as Shopping, Addresses, and Services, while low-population regions tend to expand toward Transit and Scenery categories of POI. These patterns align with the typical functional demands and regional types related to population.
> >
> > 2. House Price patterns
> >
> >     Regions with both high house prices and dense POI distributions tend to expand toward Automobile Repair, Automobile Services, and Business Residence categories. This pattern reflects their concentration of higher-value assets and more service-oriented residential environments. In contrast, regions with lower house prices and relatively sparse POI availability tend to expand toward Public Facilities and Motorcycle Services, which aligns with their more basic service needs and less intensive commercial development.
> >
> > 3. GDP patterns
> >
> >    For regions with both high GDP density and high POI density, the POI categories such as Company & Enterprise and Shopping are preferred, which is consistent with their strong economic activity and commercial intensity. Meanwhile, low-GDP regions with fewer POIs tend to expand toward Scenery and Public Facilities, reflecting their more limited industrial structure and reliance on general-purpose or recreational services.
> >
> > We appreciate your insightful suggestion for summarizing broader expansion patterns, and we will try to add more observations of spatial and semantic patterns to the appendix.
> >
> > ### Q3. House Price and GDP Density Prediction (SG and NYC)
> >
> > Thanks again for your interest in our experimental results! While the public, fine-grained House Price and GDP data for Singapore and NYC are not available, we found an estimated Singapore GDP data based on nighttime light [1]. Taking this as the ground truth, we report the performance of several competitive methods.
> >
> > - Results of GDP estimation in Singapore
> >
> > | Baseline | MAE (mean ± std)    | RMSE (mean ± std)     | R² (mean ± std)     |
> > |----------|-----------------------|-------------------------|-----------------------|
> > | BERT     | 565.09 ± 17.67        | 918.74 ± 107.89         | 0.21 ± 0.06           |
> > | OpenAI   | 576.35 ± 15.51        | 909.58 ± 104.75         | 0.22 ± 0.04           |
> > | CityFM   | 561.99 ± 19.78        | 890.61 ± 116.28         | 0.26 ± 0.05           |
> > | **SubUrban** | **559.99 ± 22.63**        | **836.35 ± 159.53**        | **0.27 ± 0.04**           |
> >
> > As reported above, SubUrban still achieves the best performance, confirming its cross-task generalization capability. We will update the full results in the revised manuscript.
> >
> > [1] M. Kummu, M. Kosonen & S. Masoumzadeh Sayyar. Downscaled gridded global dataset for gross domestic product (GDP) per capita PPP over 1990–2022. Scientific Data, 2025.

---

### Official Review · Reviewer_D3YD · 2025-11-01

**Soundness:** 2
**Presentation:** 2
**Contribution:** 2
**Rating:** 4
**Confidence:** 4

**Summary:**

This paper proposes an urban representation learning method named SubUrban aiming to reduce human efforts in feature selection and engineering. The core idea is to represent an urban region with a set of POIs within or near the region. A reinforcement learning-based approach is presented to automatically learn the representative POIs for each region (and hence the representation/embeddings). LLMs are applied to help pre-select a subset of the POIs within a region, and to guide the optimization of POI category weighting. Experimental results using data from four cities (Beijing, Shanghai, Singapore, and NYC) across three downstream tasks (Population density, house price, and GDP density prediction) showed the effectiveness of the proposed method.

**Strengths:**

1. The proposed method uses POI data only and helps avoid manual feature selections.

2. Datasets from different cities (and countries) and different downstream tasks showed the effectiveness of the proposed method.

3. Source code has been made available.

**Weaknesses:**

1. Motivation:

- The motivation of using POIs within a region and its $\delta$-neighborhood to represent the region needs further discussion and justification. Also, how is $\delta$ determined?

-  Using LLMs to generate keywords for each region to serve as POI filters seems quite restrictive (especially for less known/small regions). The LLM prompt template shown in Appendix C.1 treats each borough of NYC as a region which does not match the number of regions in NYC as shown in Table 1. It is unclear how exactly the POI pre-selection prompt is designed for each city or region. Both the motivation and implementation need further discussion.

2. Technical details:

- More details are needed on how k-means is applied to prune the POIs and why this help "regulate spatial density and ensure more uniform coverage across the regions".

- What are $q_c$ and $C$ in Equation 3?

- Where do the candidate $p_i$'s in Equation 4 come from?

- What does the prompt look like for the LLM-instructed CEM tuning process?

3. Experiments:

- The choice of baselines in the experiments needs further justification. Only two baselines are on urban representation learning. More baselines are needed:

    Li et al. Urban region representation learning with OpenStreetMap building footprints. In KDD 2023.

    Yan et al. UrbanCLIP: Learning text-enhanced urban region profiling with contrastive language-image pretraining from the web. In WWW 2024.

    Jin et al. Urban region pre-training and prompting: A graph-based approach. In KDD 2025.

    Hao et al. UrbanVLP: Multi-granularity vision-language pretraining for urban socioeconomic indicator prediction. In AAAI 2025.

While these methods may use more features, using POI only but with substantial performance gaps may not fully justify the advantage of the proposed solution.

- The population density prediction results reported in Table 2 for CityFM are close to those in Table 7 of the CityFM paper for Singapore but quite different for NYC. Clarification is needed.

- How are the LLMs and prompts chosen for the implementation? How are their choices impact overall model performance?

- It is also a bit odd to use Random Forest as the downstream task prediction model given that the downstream tasks are regression tasks.

**Questions:**

See the Weaknesses section.

---

> ### Author Response · Authors · 2025-11-26
>
> We sincerely thank you for the careful and thoughtful review. And we appreciate your recognition of our main idea and the release of the codes. We now respond to the questions you raised.
>
> ### W1. Motivation
>
> - **Justification of δ-neighborhood POI**:
>
>     Existing studies across urban analytics consistently show that incorporating POIs from surrounding area of the region benefits downstream urban tasks. Beyond Tobler’s First Law [1], empirical evidence from neighborhood based models (e.g., Urban2Vec [2]), high-order relational structures (e.g., GeoHG [3] and HyperRegion [4]), and POI-cluster hypergraph approaches for citywide demand prediction [5] collectively supports using contextual POIs from the δ-neighborhood. These findings motivate our design.
>
> - **Determination of δ**:
>
>     δ is dynamically obtained in the hard selection stage in Section 4.2.2. After soft selection provides at most 𝐾soft = 1.5𝐾 POI candidates, we compute each candidate’s cosine similarity to the region through the embedding space, and then use the mean similarity as the threshold. The POIs above this mean threshold are kept as the final set to expand. The number of expanded POIs is the automatically determined δ for each region.
>
> - **Keywords generation for less-known / small regions**:
>
>     Thank you for raising this insightful point!
>
>     - We acknowledge that for small or less-known regions, the LLM-generated keywords are often incomplete, and this limitation indeed exists in practice. Although we explicitly prompt the LLM to produce broad and diverse keyword sets, such prompting can only alleviate but not fully eliminate the issue.
>     - However, these regions typically exhibit sparse and highly irregular POI distributions, which consist of less statistically significant or recurring patterns. Even when we intentionally relax the filtering criteria to retain more POIs, the overall performance shows negligible improvement. This indicates that the additional POIs, while not meaningless, contribute little to uncovering robust statistical regularities. Furthermore, the hypernode expansion stage helps mitigate this limitation by propagating contextual signals from better-characterized regions to neighboring sparse ones.
>     - For future work, we plan to introduce a lightweight feedback mechanism that automatically identifies regions with very few filtered POIs, supplements them with a small random POI subset, and reprocesses them through the LLM to refine their keyword coverage.
>
> - **Rationale of the mismatch in the number of regions**:
>
>     We appreciate the reviewer for pointing out this issue and apologize for the confusion. The mismatch results from using different spatial granularities for different purposes. For keyword generation, we adopt a coarser and semantically meaningful administrative division (for example, NYC boroughs), which aligns better with the LLM’s knowledge. In contrast, Table 1 reports the finer regions derived from road-network segmentation following common evaluation protocols [6] [8]. These two partitions serve distinct roles and are not conflicting. In the revised version, we will explicitly differentiate the terminology by using *administrative division* for the LLM keyword extraction stage and *region* for downstream evaluation units to avoid ambiguity.
>
> - **Motivation of prompt for generating regional keyword**:
>
>     Thank you for the question. As described in Section 4.1 and Appendix C.1 that rather than feeding large and noisy POI lists to the LLM, we provide only the name of *administrative division* and a general instruction, together with a non-exhaustive set of illustrative keyword types in our prompt. This unified template provides appropriately scoped contextual guidance while remaining fully city-agnostic.
>
> - **Template of prompt for generating regional keyword**
>
>     The detailed template of the prompt for POI pre-selection in NYC is as follows:
>
>     ```text
>     For the following NYC borough: <borough_name>
>
>     Please generate a concise set of representative keywords that capture the essential characteristics and features of this borough. The following categories are provided solely as non-exhaustive illustrative examples to guide the generation of relevant keywords:
>     - Notable landmarks, buildings, or attractions (e.g., museums, parks, iconic buildings)
>     - Shopping centers, markets, or commercial districts
>     - Transportation hubs (subway stations, bridges, major streets)
>     - Cultural institutions or entertainment venues
>     - Residential developments, housing projects, or neighborhood characteristics
>     - Local businesses, restaurants, or community features
>     - Historical sites or points of interest
>     - Major neighborhoods or districts within the borough
>
>     Provide the keywords in a comma-separated format within single quotes, such as: 'keyword1','keyword2','keyword3',...
>     ```

---

> ### Author Response · Authors · 2025-11-26
>
> ### W2. Technical details
>
> - **Details of K-means pruning for POIs:**
>
>     We thank the reviewer for the request for clarification. As described in Section 4.1, we apply K-means within each region to alleviate excessive local clustering of POIs retrieved by BM25 using LLM-generated keywords. Specifically, K-means groups nearby POIs into spatial clusters, and we compute the median pairwise distance within each cluster as a threshold to define its dense core. POIs inside the core are randomly subsampled to match the density of the surrounding area, while those outside are retained. This process prevents POIs from concentrating excessively in a few small areas and yields a more balanced spatial distribution within each region. We agree that “uniform coverage” may have been an imprecise term; our goal is to regulate local spatial density rather than to enforce global uniformity.
>
> -  **𝑞𝑐 and 𝐶 in Equation 3**:
>
>     Thanks for pointing out the ambiguity of these variables. In Equation 3, 𝐶 denotes the total number of POI categories, and 𝑞𝑐 is the proportion of selected POIs belonging to category 𝑐. We will add these descriptions in the revised version.
>
> - **𝑝𝑖 in Equation 4**:
>
>     We thank the reviewer for the comment. In Equation 4, 𝑝𝑖 refers to the embedding of a candidate POI from the buffer set 𝐵𝑟, while 𝑝𝑗 denotes the embeddings of POIs that already belong to the region’s intra-region set 𝑆𝑟. We will make these definitions clear in the revised version.
>
> - **Template of prompt for generating instructions on CEM process**
>
>     An example of the detailed prompt for instructing CEM process is as follows:
>
>     ```text
>     Analyze the CEM optimization process and provide improvement suggestions.
>
>     Important Background: The current system uses a triple-task mixed reward for optimization, where mixed reward = Population prediction task R² * weight + Housing price prediction task R² * weight + GDP prediction task R² * weight. All "rewards" and "performance" metrics refer to this mixed reward value.
>
>     Current 3-round optimization summary:
>     {current_summary}
>
>     Global optimization history summary:
>     {limited_history}
>
>     Please provide the following content:
>     1. Analysis of the current triple-task mixed reward optimization state, particularly focusing on whether local optimum problems exist
>     2. Identify which POI categories significantly affect triple-task mixed performance (positive or negative)
>     3. Specific suggestions on how to adjust CEM parameters:
>     - For categories with the greatest weight impact, suggest significant adjustments (±0.5 or more)
>     - For categories with moderate weight impact, suggest moderate adjustments (±0.2 to ±0.4)
>     - Whether smoothing_factor needs adjustment, considering more aggressive exploration strategies
>     - Whether elite_fraction needs adjustment
>     - Provide a larger standard deviation (0.2-0.5) for specific categories to increase exploration
>     4. If optimization stagnates, suggest restarting distribution parameters for at least 3 categories
>
>     Please provide specific parameter adjustment suggestions in JSON format as follows:
>     {
>     "category_adjustments": [
>         {"name": "category_name", "mean_adjustment": 0.5, "std_adjustment": 0.3}
>     ],
>     "global_adjustments": {
>         "smoothing_factor": 0.1,
>         "elite_fraction": 0.05
>     },
>     "restart_categories": ["category1", "category2", "category3"]
>     }
>     ```

---

> ### Author Response · Authors · 2025-11-26
>
> ### W3. Experiments
>
> - **Results of more URL baselines:**
>
>     We thank the reviewer for the valuable suggestions. We are currently trying to incorporate the listed methods into our pipeline. We give the results of three tasks by UrbanCLIP in **Beijing**, and we also put the performance of our SubUrban here for comparison.
>
>     | Baseline   | POP-MAE | POP-RMSE | POP-R² | HOUSE-MAE | HOUSE-RMSE | HOUSE-R² | GDP-MAE | GDP-RMSE | GDP-R² |
>     |------------|---------|----------|--------|-----------|------------|----------|---------|----------|--------|
>     | UrbanCLIP  | 5691.76 | 8571.37  | 0.42 | 21714.80  | 30545.68   | 0.44   | 949.04  | 1329.87  | -0.09 |
>     | SubUrban | 3283.11 | 5719.89 | 0.72 | 12235.97 | 17021.29 | 0.85 | 349.63 | 568.85 | 0.80 |
>
>     These results show that although UrbanCLIP leverages both remote-sensing imagery and textual descriptions, its downstream performance still does not exceed that of our POI-only SubUrban.
>
>     For the results of more multimodal fusion baselines, we will update immediately once the experiments are completed.
>
> - **Clarification on Population Density Prediction results in SG and NYC**
>
>     We apologize for the scale differences of the ground truth between Singapore and other cities. The reason why our results for Singapore in Table 2 are close to the results of Table 7 in CityFM [6] is that we follow the unit 'thousands of people per square kilometer', while for the rest of the cities we use the unit 'people per square kilometer'. A concise summary of the results after we unified the dimensions of ground truth in Singapore is as follows:
>
>     | Model        | MAE (±std) ↓           | RMSE (±std) ↓          | R² (±std) ↑        |
>     | ------------ | --------------------- | --------------------- | ---------------- |
>     | BERT-Avg     | 4002.01 (±206.71)     | 5818.71 (±329.95)     | 0.68 (±0.01)     |
>     | OpenAI-Avg   | 3896.15 (±85.61)      | 5657.27 (±89.50)      | 0.69 (±0.03)     |
>     | GraphSage    | 3424.71 (±117.11)     | 5280.75 (±206.49)     | 0.74 (±0.02)     |
>     | DGI          | 3925.79 (±206.24)     | 5720.04 (±385.50)     | 0.73 (±0.03)     |
>     | MVGRL        | 4014.24 (±301.56)     | 5932.78 (±542.86)     | 0.70 (±0.03)     |
>     | HGI          | 3393.52 (±216.56)     | 5035.43 (±295.80)     | 0.76 (±0.02)     |
>     | CityFM       | 3085.52 (±104.42)     | 4504.32 (±203.52)     | 0.82 (±0.01)     |
>     | **SubUrban** | **2475.59 (±180.29)** | **4266.60 (±455.03)** | **0.86 (±0.03)** |
>
>     These results from Singapore will be updated to the revised PDF of our paper.
>
> - **Impact of different choices on LLM**
>
>     The core role of LLM in our framework is to provide an efficient initialization of POI inputs for RL training and to accelerate the convergence of the CEM optimization. The prompts used in two of LLM instructed steps are simple and not heavily engineered. While LLM guidance is necessary for bootstrapping the process, our experiments show that neither the specific choice of LLM nor the exact prompt design is a major factor influencing downstream performance. These components support the pipeline but are not the focus or main contribution of our method. We report the CEM iteration statistics for different LLMs to illustrate how LLM choice influences the CEM optimization process, which is computed over five repeated runs in Beijing as below:
>
>     | LLM Type | Initial Reward (avg) | Final Reward (avg) | End Iteration |
>     |-------------|-----------------------|---------------------|----|
>     | No LLM      | 0.4586                | 0.5272              | 13 |
>     | DeepSeek-R1 | 0.4855                | 0.5490              | 9  |
>     | GPT-3.5     | 0.4770                | 0.5796              | 11 |
>     | GPT-4       | 0.5033                | 0.5532              | 8  |
>
>     The results show that GPT-4 offers the strongest optimization gains, while GPT-3.5 and DeepSeek-R1 deliver solid improvements at lower cost.

---

> ### Author Response · Authors · 2025-11-26
>
> - **Impact of different prediction models for evaluation**
>
> Random Forest (RF) is widely adopted as an evaluator in urban and geospatial representation learning, largely due to its strong nonlinear modeling capacity and robustness to heterogeneous or noisy regional features [7]. This usage is consistent with prior work in region embedding and spatial prediction [8,9,10]. To further prove our choice of predictor, the results of **Beijing** using Linear Regression, Multilayer Perceptron, and Random Forest are shown below:
>
> | Predictor | Baseline  | Pop-MAE     | Pop-R²   | House-MAE    | House-R²    | GDP-MAE    | GDP-R²   |
> | --------- | --------- | ----------- | -------- | ------------ | ----------- | ---------- | -------- |
> | LR        | BERT      | 14671.41    | < -1   | 20279.71     | 0.57        | 1250.43    | < -1    |
> | LR        | OpenAI    | 16735.80    | < -1   | 17404.57     | 0.60        | 1333.68    | < -1    |
> | LR        | DGI       | 14663.68    | < -1   | 39151.84     | -0.86       | 1185.24    | < -1    |
> | LR        | MVGRL     | –           | –        | –            | –           | –          | –        |
> | LR        | GraphSage | 55415.32    | < -1   | 16871.31 | 0.62    | 4702.95    | < -1  |
> | LR        | HGI       | 6244.53 | 0.33 | 46755.54     | < -1       | 603.31 | 0.51 |
> | LR        | CityFM    | 43973.64    | < -1 | > 10^6  | < -1 | 3654.18    | < -1 |
> | LR        | SubUrban  | 9956.67     | -0.70    | 22208.87     |   0.45        | 922.45     | -0.08    |
> | --------- | --------- | ----------- | -------- | ------------ | -------- | ---------- | -------- |
> | MLP       | BERT      | 4462.83     | 0.52     | 25106.37     | 0.25     | 430.23     | 0.65     |
> | MLP       | OpenAI    | 4468.77     | 0.52     | 23772.01     | 0.37     | 423.81     | 0.65     |
> | MLP       | DGI       | 4599.54     | 0.47     | 29267.81     | 0.03     | 421.50     | 0.67     |
> | MLP       | MVGRL     | 5507.85     | 0.27     | 28000.79     | 0.14     | 521.95     | 0.45     |
> | MLP       | GraphSage | 4185.67     | 0.56     | 14535.06     | 0.78     | 425.49     | 0.68     |
> | MLP       | HGI       | 6157.84     | 0.33     | 38241.42     | < -1     | 571.67     | 0.52     |
> | MLP       | CityFM    | 3943.63     | 0.62     | 18618.74     | 0.66     |   336.48   | 0.81     |
> | MLP       | SubUrban  | 3793.28 | 0.67 | 14122.74 | 0.79 | **331.82** | **0.83** |
> | --------- | --------- | ------- | ------ | --------- | -------- | ------- | ------ |
> | RF        | BERT      | 5043.73 | 0.49   | 14391.39  | 0.74     | 490.47  | 0.62   |
> | RF        | OpenAI    | 5419.69 | 0.46   | 13946.38  | 0.75     | 523.66  | 0.59   |
> | RF        | DGI       | 4990.86 | 0.47   | 15357.90  | 0.75     | 466.77  | 0.67   |
> | RF        | MVGRL     | 4990.86 | 0.47   | 15692.40  | 0.70     | 502.90  | 0.57   |
> | RF        | GraphSage | 4774.99 | 0.52   | 14748.74  | 0.69     | 488.91  | 0.63   |
> | RF        | HGI       | 4534.83 | 0.56   | 14719.13  | 0.78     | 409.07  | 0.70   |
> | RF        | CityFM    | 4199.19 | 0.64   | 14291.54  | 0.75     | 384.27  | 0.78   |
> | RF        | **SubUrban**  | **3283.11** | **0.72**   | **12235.97**  | **0.85**     | 349.63  | 0.80   |
>
> **SubUrban achieves the best performance on all tasks with both RF and MLP predictors.** However, while most of the baselines perform worse with LR (e.g., MVGRL exhibits numerical instability when fitted with LR) since these urban socioeconomic regression tasks involve strong nonlinear dependencies. Meanwhile, some baselines do not perform stably with the MLP predictor. In this case, we take the results of RF into our paper since all of the baselines perform well and are stable with this predictor.
>
> ### **References**
>
> [1] Tobler, W. R. A Computer Movie Simulating Urban Growth in the Detroit Region. Economic Geography, 1970.
>
> [2] Lewis, P. et al. Retrieval-Augmented Generation for Knowledge-Intensive NLP Tasks. NeurIPS, 2020.
>
> [3] Zou, X. et al. Learning Geospatial Region Embedding with Heterogeneous Graph (GeoHG). arXiv, 2024.
>
> [4] Deng, Z. et al. HyperRegion: Integrating Graph and Hypergraph Contrastive Learning for Region Embeddings. IEEE Transactions on Mobile Computing, 2025.
>
> [5] Kuang, Y. et al. Attention-based Citywide Electric Vehicle Charging Demand Prediction Approach Considering Urban Region and Dynamic Influences. arXiv, 2024.
>
> [6] Pasquale B. et al. City foundation models for learning general purpose representations from openstreetmap. CIKM, 2024.
>
> [7] Breiman, L. Random Forests. Machine Learning, 2001.
>
> [8] Yi Li. et al. Urban region representation learning with openstreetmap building footprints. KDD, 2023.
>
> [9] Loddi, G., et al. Urban Region Embeddings from Service-Specific Mobile Traffic Data. IEEE MDM, 2025.
>
> [10] Georganos, S., Grippa, T., Lennert, M. Geographical Random Forests: A Spatial Extension of the Random Forest Algorithm to Address Spatial Heterogeneity in Remote Sensing and Population Modelling. IJGIS, 2021.

---

### Author Response · Authors · 2025-12-03
**General Response**

We sincerely appreciate the reviewers’ thoughtful comments and the time they dedicated to evaluating our work.

We are encouraged that the reviewers recognize SubUrban as an autonomous, data-efficient, and broadly generalizable framework for urban representation learning. By **submodular-driven reinforcement learning with LLM instructions for POI-based hypernode expansion**, our approach proves effective across diverse cities and tasks while achieving stronger data efficiency than methods that rely on handcrafted features or heavy model-specific tuning.

Overall, the reviews addressed the following strengths:

- **Conceptual Clarity**: Reviewers noted that SubUrban’s core formulation is clear and intuitive. Modeling POI expansion as a submodular sequential process with Coverage, Saturation, and Buffer as states provides a transparent and interpretable structure for learning regional representations (highlighted by reviewers mxB9, oxjG, t4iD).

- **Autonomous Utility**: Reviewers noted that SubUrban reduces manual feature engineering and city-specific heuristics by autonomously selecting informative POIs, providing a general framework that remains effective across different cities and downstream tasks (highlighted by reviewers D3YD, mxB9, oxjG).

- **Efficiency**: Reviewers pointed out the use of LLMs as an interesting method. LLM assists in semantically prefiltering POIs and provides iterative guidance that accelerates the convergence of CEM optimization, which forms an effective LLM-in-the-loop mechanism that improves the efficiency of this framework (highlighted by reviewers mxB9, oxjG).

To address the reviewers' concerns, we incorporated additional clarifications and experiments into the revised manuscript, including:

- **Clarification of Design Details**

    We clarified the motivation and dynamic computation of δ-neighborhood POIs, distinguished the differently defined region granularities used for keyword generation and downstream evaluation, added missing descriptions (for 𝑞𝑐, C, 𝑝𝑖), detailed the K-means pruning procedure, and explained the conceptual roles of α and Top-K as search-control parameters rather than heuristics. (Addressing reviewers D3YD W1-W2, mxB9 W2, t4iD W2; updated Section 4.1-4.2, and Table 2 in Section 5.2)

- **LLM Usage Analysis**

    We performed ablations comparing LLM preprocessing with Information Gain and random sampling, reported detailed LLM call/cost/convergence statistics across multiple LLM instructed steps, evaluated reproducibility of keyword generation, and detailed prompt templates. (Addressing reviewers mxB9 W2-W3, oxjG W3, t4iD W1; added in Appendix C, and Appendix D.10)

- **Reward Behavior Analysis**

    We added per-module reward trajectories and marginal-gain statistics. (Addressing reviewers mxB9 Q1, oxjG W3-W4; added in Appendix D.4 and D.8)

- **Extended Comparison**

    We conducted additional robustness checks under unified embedding dimensions, different evaluators (LR/MLP/RF), and grid-based region partitions. And we compared with a multimodal baseline UrbanCLIP, evaluated GDP density in Singapore using a nighttime-light–derived dataset. (Addressing reviewers D3YD W3, mxB9 W1 and Q3; added in Appendix D.3, D.5-D.7, and D.9)

All revisions in the paper are marked in blue.

We extend our heartfelt thanks to all reviewers for their constructive feedback and recognition of our work. We welcome further discussion and look forward to continued engagement. Thank you!

---

### Meta-Review · Area_Chair_DeCM · 2025-12-24

**Summary:**

All reviewers acknowledged that the proposed SubUrban framework, which utilizes reinforcement learning and LLMs for autonomous urban region representation, is a meaningful attempt to reduce reliance on manual feature engineering. However, they expressed concerns regarding the system's complexity, the fairness of the experimental comparisons, and the necessity of the LLM components, and all reviewers assigned scores that do not support acceptance.

Reviewer D3YD and Reviewer mxB9 both raised concerns about the over-engineering of the system, questioning whether the complexity of combining RL, submodularity, and LLMs is justified by the performance gains given the computational costs. Reviewer oxjG pointed out significant flaws in the experimental design, specifically regarding inconsistent embedding dimensions across baselines and the choice of evaluators. Reviewers t4iD and mxB9 questioned the necessity and reproducibility of the LLM components, suggesting that simpler statistical methods (like Information Gain) might suffice or that the LLM introduces instability in feature generation.

**Reviewer Concerns:**

The authors responded by adding extensive supplementary experiments in the Appendix, including an efficiency analysis to address concerns about computational and economic costs , and fairness comparisons with unified embedding dimensions and grid partitions to answer questions regarding experimental rigor. They also provided ablation studies to justify the contribution of the LLM and RL components. However, these revisions and explanations did not appear sufficient to address the reviewers’ fundamental concerns regarding the over-engineering of the framework and the necessity of such a complex pipeline. The reliance on closed-source LLMs (e.g., GPT-4) for feature preprocessing still raises outstanding issues regarding reproducibility and practical deployment efficiency that cannot be fully resolved by cost tables alone.

**Reviewer Scores:**

As none of the reviewers continued to participate in the discussion after the rebuttal, I believe that the original scores should remain unchanged. Even acknowledging the controversy regarding one reviewer's score, the overall ratings remain insufficient to warrant acceptance.

---

### Decision · Program_Chairs · 2026-01-26

Reject